



# Snowpack dynamics in the Lebanese mountains from quasi-dynamically downscaled ERA5 reanalysis updated by assimilating remotely-sensed fractional snow-covered area

Esteban Alonso-González[1], Ethan Gutmann[2], Kristoffer Aalstad[3], Abbas Fayad[4], Simon Gascoin[5]

1- Instituto Pirenaico de Ecología, Spanish Research Council (IPE-CSIC), Zaragoza, Spain
2- Research Application Laboratory, National Center for Atmospheric Research (RAL-NCAR), Boulder, CO, United States
3- Department of Geosciences, University of Oslo, Oslo, Norway
4- Centre for Hydrology, University of Saskatchewan, Saskatoon, Saskatchewan, Canada
5- Centre d'Etudes Spatiales de la Biosphère (CESBIO), UPS/CNRS/IRD/INRA/CNES, Toulouse, France

**Abstract:** The snowpack over the Mediterranean mountains constitutes a key water resource for the downstream populations. However, its dynamics have not been studied in detail yet in many areas, mostly because of the scarcity of snowpack observations. In this work, we present a characterization of the snowpack over the two mountain ranges of Lebanon. To obtain the necessary snowpack information, we have developed a 1 km regional scale snow reanalysis (ICAR_assim) covering the period 2010-2017. ICAR_assim was developed by means of ensemble-based data assimilation of MODIS fractional snow-covered area (fSCA) through the energy and mass balance model the Flexible Snow Model (FSM2), using the Particle Batch Smoother (PBS). The meteorological forcing data was obtained by a regional atmospheric simulation developed through the Intermediate Complexity Atmospheric Research model (ICAR) nested inside a coarser regional simulation developed by the Weather Research and Forecasting model (WRF). The boundary and initial conditions of WRF were provided by the ERA5 atmospheric reanalysis. ICAR_assim showed very good agreement with MODIS gap-filled snow products, with a spatial correlation of R = 0.98 in the snow probability (P(snow)), and a temporal correlation of R = 0.88 in the day of peak snow water equivalent (SWE). Similarly, ICAR_assim has shown a correlation with the seasonal mean SWE of R = 0.75 compared with in-situ observations from Automatic Weather Stations (AWS). The results highlight the high temporal variability of the snowpack in the Lebanon ranges, with differences between Mount Lebanon and Anti-Lebanon that cannot be only explained by its hypsography been Anti-Lebanon in the rain shadow of Mount Lebanon. The maximum fresh water stored in the snowpack is in the middle elevations approximately between 2200 and 2500 m a.s.l. Thus, the resilience to further warming is low for the snow water resources of Lebanon due to the proximity of the snowpack to the zero isotherm.



**Keywords** — Snow, dynamical downscaling, data assimilation, fractional snow
cover, Mediterranean Mountains

## 1. Introduction

The hydrological processes related to mountain areas are essential for the water supplies to a large part of humanity (Viviroli et al., 2007). Despite the relatively mild temperature of the Mediterranean climates, mountains there often exhibits deep and long lasting snowpacks (Alonso-González et al., 2020; Fayad et al., 2017b). Thus, as most of the annual precipitations falls during winter season (García-Ruiz et al., 2011) the mountain snowpack strongly reshapes the hydrographs to sustaine high flows until the end of the spring (López-Moreno and García-Ruiz 2004), permitting better synchronization of water demand and availability during the dry season (García-Ruiz et al., 2011). Mediterranean snowpacks are characterized by a high interannual variability. Despite this variability, the thickness and high density exhibited by Mediterranean snowpacks (Fayad et al., 2017b), makes them an effective water storage system. In addition, high sublimation rates are associated with Mediterranean snowpacks (Fayad and Gascoin, 2020; Herrero et al., 2016; Schulz and de Jong, 2004). The fact that snowpack conditions are close to isothermal during most of the snow season makes them highly sensitive to the current climate warming (Alonso-González et al., 2020a; López-Moreno et al., 2017; Yilmaz et al., 2019).

The Lebanon Mountains are a clear example of Mediterranean mountains, where snow exerts a key control on the hydrology and water resources are critically dependent on the interannual fluctuations of the snow pack (El-Fadel et al., 2000). Despite its importance, snow observations in the region are scarce (Fayad et al., 2017a), making the study of distributed snow dynamics challenging. Recently, Fayad and Gascoin (2020) have develop distributed snowpack simulations over key areas of Mount Lebanon, forcing the model by interpolating observations of the few existing automatic weather stations (AWS) using the SnowModel by Liston and Elder (2006). They showed the importance of the liquid water percolation scheme given the isothermal condition of the snowpack and estimated the snow water equivalent over three key catchments in the windward western divide of Mount Lebanon. However, due to the lack of meteorological data outside this area, these simulations did not cover the whole mountain area of the country and were limited to three snow seasons.

Remote sensing and numerical modeling have become reliable tools to generate useful meteorological information for mountain regions (Lundquist et al., 2019), and also to generate robust snow data worldwide. Atmospheric reanalyses are a valuable source of long term (multidecadal) climatological information, especially at





planetary scales (e.g. Wegmann et al., 2017; Wu et al., 2018). However, spatially downscaling such products is mandatory to derive relevant snow information over complex terrain (Baba et al., 2018b; Mernild et al., 2017 among others). Dynamical downscaling has been shown to outperform statistically gridded products for meteorological variables in complex terrain (Gutmann et al., 2012). More specifically, high resolution fully dynamical meteorological models can reproduce the snowfall patterns over complex terrain (Ikeda et al., 2010; Rasmussen et al., 2011). However, the computational cost of fully dynamical downscaling solutions becomes prohibitive for large domains at high spatial resolutions. To reduce the computational cost, many different solutions of varying complexity have been developed using statistical interpolations corrected with the topography or using simplifications of the atmospheric dynamics (Fiddes and Gruber, 2014; Gutmann et al., 2016; Liston and Elder, 2006). In this way, energy and mass balance snowpack models have been coupled with atmospheric models to develop multidecadal snow simulations (Alonso-González et al., 2018; van Pelt et al., 2016 among others). In addition, remote sensing products have been widely used to study the duration and variability of the snow cover (Gascoin et al., 2015; Saavedra et al., 2017; Yilmaz et al., 2019). However, less often, numerical modeling and remote sensing have been combined in a data assimilation framework to study the multiyear snowpack dynamics. Assimilation of remoted sensed snow cover observations has been shown to improve numerical snowpack models outputs in both distributed (e.g. Baba et al., 2018; Margulis et al., 2016) and semi distributed simulations (Cluzet et al., 2020; Fiddes et al., 2019). These approaches are particularly promising in data-scarce regions to reduce the biases in atmospheric forcing.

In this work, we have simulated the snowpack of the Lebanon Mountains, as an alternative to sparse snowpack observations .We have generated a 1 km resolution snowpack reanalysis, using an ensemble based assimilation of fractional snow cover area (fSCA) obtained from the Moderate Resolution Imaging Spectroradiometer (MODIS) satellite sensor. More specifically, the ERA5 reanalysis  (Hersbach, 2016) was dynamically downscaled using regional atmospheric models in two steps. First, a 10 km resolution atmospheric simulation using the Weather Research and Forecast model (WRF) (Skamarock et al., 2008) was performed covering the period between 2010 and 2017. Then, a finer 1 km simulation using the Intermediate Complexity Atmospheric Research model (ICAR) (Gutmann et al., 2016) was nested inside the previous WRF simulation covering the time period. To improve the ICAR snowpack outputs, the new meteorological data generated was used to force an energy and mass balance snowpack model, the Flexible Snow Model (FSM2) (Essery, 2015), previously perturbing the meteorological fields to generate an ensemble of snowpack simulations. Then, the Particle Batch Smother (PBS) (Margulis et al., 2015), a Bayesian data assimilation scheme, was applied to assimilate daily remotely sensed fSCA information. We tested the generated snow products in the mountains of Lebanon with independent observations. Finally, the dynamics of the



snowpack in the mountains of Lebanon are studied from the generated multi-year snow time series.

## 2. Study area

Lebanon is a country located on the eastern Mediterranean Sea between latitudes 33° and 35° N. Its climatology is typical Mediterranean influenced mainly by its

proximity to the Mediterranean Sea and its complex topography (Figure 1). There are two main mountain ranges that run in parallel to the Mediterranean coast from North to South. These mountain ranges are the Mount Lebanon and Anti-Lebanon Mountains, reaching 3088 m a.s.l. (Qurnat as Sawdā peak) and 2814 m a.s.l. (Mount Hermon peak) respectively.

Despite Lebanon having more available water resources than its neighboring countries, it is considered a water scarce region (El-Fadel et al., 2000), where droughts are frequent and are expected to increase due to climate change (Farajalla et al., 2011). The particularl distribution of its mountain ranges constitutes an effective topographical barrier to humidity advected from the Mediterranean sea,

enhancing the winter precipitation as consequence of orographic effects (Jomaa et al., 2019). Thus, a seasonal snowpack appears every year lying over a wide area of the country (Mhawej et al., 2014).

It was estimated from satellite retrievals of snow cover that 31% of the spring discharge of Lebanon is associated with snow melt (Telesca et al., 2014). In addition,

the groundwater dynamics of Lebanon are mainly controlled by the snow melt as consequence of its karstic nature (Bakalowicz et al., 2008; El-Fadel et al., 2000). Thus, the water resource provided by the snowpack is crucial for the Lebanese society. The dependence of Lebanon on snow resources became more acute during the recent drought in the Eastern Mediterranean (Cook et al., 2016). In addition, the

water stress increased notably in recent years partially due to the increase in domestic water demand, agricultural water use, and the Syrian refugee crisis (Jaafar et al., 2020) but also to the poor management of the water resources, and water pollution.



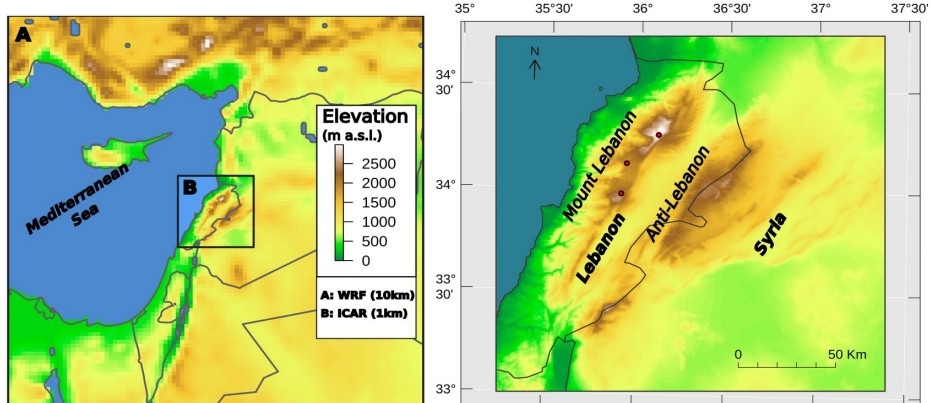

*Figure 1: Atmospheric models domain configuration (left) and Lebanon Localization map (right). The red dots represent the AWS positions.*

## 3. Data and Methods

### 3.1 Regional atmospheric simulations configuration

To generate the meteorological forcing, we used the ICAR atmospheric model nested inside a WRF simulation forced by the ERA5 reanalysis. Previously, the WRF model was used to generate a regional atmospheric simulation on a 10 km x 10 km grid, covering the eastern part of the Mediterranean Sea with 100 x 100 grid cells, centered over Lebanon's Mountains (Figure 1). In the vertical dimension, the domain is composed of 35 levels with the top set to 50 hPa. The simulation covers the period from 01$^{st}$ of January 2010 to 30$^{th}$ of June 2017, using the first 9 months as spin-up period allowing for physical equilibrium between the external forgings and the land model (Montavez et al., 2017). We used the ERA5 reanalysis dataset at an hourly frequency as boundary and initial conditions of the WRF model. The ERA5 dataset is an atmospheric reanalysis, which replaces the widely used ERA-Interim reanalysis (Berrisford et al., 2009). It has a spatial grid of 30 km with 138 vertical levels with the top at 80 km. It proved to out perform ERA-interim in many climatological applications and as a forcing dataset for different modeling applications (Albergel et al., 2018; Tarek et al., 2019; Wang et al., 2019 among others). The parametrization schemes used in the WRF simulation include: the Thompson cloud microphysics scheme (Thompson et al., 2008), the NCAR Community Atmosphere Model (CAM) scheme for both shortwave and longwave radiations (Neale et al., 2004), the Noah-MP scheme for the land surface physics (Niu et al., 2011), the Mellor-Yamada-Janjic scheme for the planetary boundary layer



(Janjic, 2002) and the Betts-Miller-Janjic scheme (Betts and Miller, 1986; Janjic, 1994) for deep and shallow convection. This WRF configuration has proved its consistency in previous studies simulating the seasonal snowpack over complex

terrain (Ikeda et al., 2010; Rasmussen et al., 2011). In addition to the described parametrization, we applied the spectral nudging technique to satisfy the large scale atmospheric conditions at the higher altitudes, while allowing the model to have its own dynamics inside the planetary boundary layer (Von Storch et al., 2000; Waldron et al., 1996). The spectral nudging technique was applied for the wind

vectors, temperature and geopotential with a wave number of 1 in each direction, based on the parameters recommended by Gómez and Miguez-Macho (2017), and nudging the waves above ~ 1000 km wavelength.

Next, the ICAR model was used to obtain a finer 1 km x 1 km spatial grid atmospheric simulation nested in the aforementioned WRF simulation domain. This

enabled us to significantly reduce the high computational cost compared to a long-term high-resolution WRF simulation. ICAR is a 4D mesoatmospheric model designed for downscaling purposes based on linear mountain wave theory. The linear theory allows ICAR to compute the main dynamical effect of topography on the atmosphere using an analytical solution, thus avoiding the need to solve the

Navier-Stokes equations and reducing computational cost by a factor of 100. The center of the ICAR simulation was established in the center of the WRF simulation, using 179 x 179 grid cells in both latitude and longitude directions and preventing the boundaries from intersecting complex terrain. The model top was situated at 4150 m above the topography with 12 vertical levels, using the default model levels

heights (Horak et al., 2019). The model configuration used: the Thompson cloud microphysics scheme (Thompson et al., 2008), the Noah land surface model (Chen and Dudhia, 2001) and the Multidimensional Positive Definite Advection Transport Algorithm (MPDATA) for the advection (Smolarkiewicz and Margolin, 1998). Convection schemes were not implemented for this simulation and the radiative

fluxes at the surface were prescribed by WRF.

## 3.2 Ensemble-based fractional snow cover assimilation

### 3.2.1 MODIS fractional snow cover area data estimation

For this study, we used satellite observations of fSCA, assimilated in an ensemble of snow simulations to improve the snow water equivalent products (SWE) of ICAR.

The daily fSCA information was obtained by means of the MODIS sensor, which is orbiting the Earth on board two satellites, Terra and Aqua. We have chosen MODIS because of its daily revisit time combined with a spatial resolution of 500 m, which is higher than our ICAR simulation. More specifically, we have used the normalized difference snow index (NDSI) retrievals of the collection 6 of the NASA snow-cover

products MOD10A1 (Terra) (Hall et al., 2006) and MYD10A1 (Aqua) (Hall and Riggs,





2016) distributed by the National Snow and Ice Data Center. To estimate the fSCA from the MODIS NDSI we have used a linear function following Salomonson and Appel (2004). The coefficients of the function were optimized using a series of 20 m resolution snow products from Theia (Gascoin et al., 2019). The Theia Snow collection provides snow cover area maps that were derived from Sentinel-2 observations. We downloaded 645 Theia Sentinel-2 snow products acquired between 2017-09-03 and 2018-12-24 over Lebanon. Theia binary snow maps were resampled to 500 m fSCA in the same grid as the MODIS products by averaging the contributing pixels. By comparing these fSCA Theia maps with the MOD10A1 products we could find $5.84 \times 10^4$ cloud-free pixels which corresponded to MOD10A1 snow-covered pixels on the same date. A subset of 40% of these NDSI-fSCA were used to fit a linear function using the least squares method. The optimized function was tested against the remaining data and yielded an fSCA RMSE of 11% and a mean absolute error of 5.7%. The same analysis was done with MYD10A1 (Aqua) products but we did not use them in the following as they exhibited a lower agreement with the Theia Sentinel-2 snow cover products (RMSE of 21%). The lower agreement of MYD10A1 is likely due to degraded detectors (Wang et al., 2012) but may also be related to the difference between the overpass time of Sentinel-2 (10:30 local time) and Aqua (13:30 local time).

We reprojected the generated MODIS fSCA products to the spheroid datum (6370 km earth radius) Lambert conformal projection used in the ICAR simulation. To avoid artifacts as consequence of the data gaps of MODIS imagery caused by the cloud cover, we have performed the aggregation when the majority of the MODIS cells used to calculate each new resampled cell was cloud free (less than 25% cloud cover), otherwise the cell was considered empty. In previous studies, the MODIS fSCA products have proved to have a good performance retrieving fSCA information compared with field observations even considering its moderate resolution (Aalstad et al., 2020). Thus, they are a robust resource to use when developing regional scale snow reanalysis.

### 3.2.2 Particle batch smoother implementation

The assimilation procedure was implemented using the PBS scheme (Margulis et al., 2015). The PBS assigns a weight to each ensemble member according to its agreement with the observations through Bayes theorem. The most obvious advantage of this technique is its computational efficiency, as it avoids the resampling step common in other assimilation algorithms. A complete description of the PBS can be found in Margulis et al. (2015). It is also summarized in Aalstad et al. (2018) and Fiddes et al. (2019). The PBS has been shown to perform well relative to other assimilation algorithms when used to assimilate fSCA information (Aalstad et al., 2018; Margulis et al., 2015), even though it can suffer from particle degeneracy as consequence of a highly inhomogeneous distribution of weights (Van



Leeuwen, 2009). The PBS has been successfully used to develop a series of snowpack reanalyses (Cortés et al., 2016; Fiddes et al., 2019; Margulis et al., 2016).

For the prior of the PBS implementation, we generated an ensemble of snowpack simulations forcing the FSM2 (Essery, 2015), with the ICAR predicted surface
meteorology. The configuration of the FSM2 model includes albedo correction as snow ages with time differently for melt and cold snow, and increased with snowfall with a maximum of 0.9. The compaction rate was calculated based on overburden and thermal metamorphism (Verseghy, 1991). The turbulent exchange coefficient was corrected based on the bulk Richardson number. The thermal conductivity was
calculated based on snow density. Finally, the FSM2 configuration accounted for retention and refreezing of water inside the snowpack. Such a configuration has been shown to properly simulate the inter- and intra-annual variability of the snowpack dynamics over  mountains with a similar Mediterranean climate (Alonso-González et al., 2018).

To generate the ensemble of forcing datasets, we perturbed the precipitation and the 2 m air temperature surface fields of the ICAR output using a log-normal and a normal (Gaussian) probability density functions respectively. We choose the mean of the probability functions from the averaged biases of the ICAR simulation, estimated form independent observations provided by three mountain AWS at the
locations shown in Figure 1 (Fayad et al., 2017a). The variance of the probability distribution functions was calculated increasing the variance of the errors by a factor of two to increase the spread of the ensemble to cover the uncertainty in the of ICAR outputs. The precipitation phase had to be recalculated for the new synthetic temperatures for each ensemble member. Due to the strong dependency
of the snowpack over Lebanon on precipitation phase, a simple temperature threshold based precipitation phase partitions are not recommended (Fayad and Gascoin, 2020). Instead, we have used the psychrometric energy balance method approach proposed by Harder and Pomeroy (2013), where the precipitation phase is estimated by means of the estimation of the temperature of the falling
hydrometeor calculated form the temperature and relative humidity. A total of 400 ensemble members per ICAR cell were independently generated by randomly drawing multiplicative time-constant parameters from the log-normal probability function for precipitation and additive parameters from the normal probability function for the 2 m air temperature.

To estimate the fSCA of each ensemble member we used the probabilistic snow depletion curve proposed by Liston (2004). This model simulates the subgrid peak SWE distribution using a lognormal probability density function. Then, the fSCA is diagnosed using the accumulated melt depth estimated from the energy balance outputs of the FSM2, the peak mean SWE, and the peak subgrid coefficient of
variation of the lognormal probability density function, assuming a constant melt



over the grid cell. The coefficient of variation of the lognormal probability density function used in this model is strongly controlled by the characteristics of the terrain. We have included this parameter as part of the assimilation, perturbing its value inside the recommended values in Liston (2004) using a mean of 0.4 and a variance of 0.01 (Aalstad et al., 2018). The PBS was implemented over the fSCA ensemble over each grid cell and season independently, using the values of the melting season, corresponding with the months of March through June. Finally, the generated SWE products (ICAR_assim hereafter) were estimated from the weighted mean of the SWE of the ensemble members, where the weights were obtained using the PBS.

## 3.3 Validation procedure and analysis of the SWE products

The ICAR atmospheric simulation and the ICAR_assim products were compared against independent observations. First, the ICAR atmospheric simulation was compared with three automatic weather stations (AWS) located in the main mountain range of the domain (Fayad et al., 2017a)(Figure 1). Temperature and precipitation measurements were aggregated to the hourly model output frequency from the original 30-minute time resolution. Then, the temperature and precipitation biases were estimated. The precipitation data was available only in two of the AWS. The error values and its variance were used to define the shape of the probability density functions of the perturbation parameters described above to generate each ensemble.

After the PBS implementation, we compared the ICAR and ICAR_assim snow products with the snow depth observed information of the three AWS. The observed snow depth was transformed into SWE by assuming a constant snow density value of 467 kg m$^{-3}$ estimated from observations in the area (Fayad et al., 2017a). That was necessary to make the AWS data comparable with the ICAR snow outputs as they are provided only as SWE. Even if it is commonly implemented in operational atmospheric forecast models, the assumption of a constant density could introduce obvious bias in the SWE estimation (Dawson et al., 2017). In the Mediterranean snowpacks, such biases are partially reduced as consequence of the high densification rates of the snowpack (Bormann et al., 2013; Fayad et al., 2017b). However, we introduced a sensitivity analysis in the comparison varying the density value in the range of ± 15% to illustrate such uncertainty. To compensate the big shift between the ICAR and ICAR_assim resolutions (1 km x 1 km) and the point-scale nature of the AWS observations, we have interpolated a new SWE series from the 4 nearest cells of the simulations using the inverse distance method. Then, the spatial accuracy of the SWE products was compared against satellite observations. First, we developed a daily gapfilled snow cover time series covering the time period of the ICAR simulation from the MODIS snow cover products using the methodology proposed by Gascoin et al. (2015). Then, the products were aggregated



to estimate the averaged snow presence over each cell in percentage ($P_{(snow)}$). The MODIS $P_{(snow)}$ product was aggregated to the ICAR grid to make it comparable. Then, we calculate the $P_{(snow)}$ for the ICAR and ICAR_assim simulations. We chose a SWE MODIS detection threshold of 20 mm to calculate the $P_{(snow)}$ from the simulated SWE

series, inside the range recommended by Gascoin et al. (2015). All the spatial analyses and the data assimilation was computed over the areas that had exhibited a P(snow) > 5%, which amounts to a total area of 4412 km2.

## 4. Results and Discussion

### 4.1 Atmospheric simulation results

The use of ICAR is justified as it is computationally inexpensive compared to similar WRF simulations, while retaining a physical basis to enable simulations in regions lacking observations. The speed up factors can range from 140 in its more complex configurations (as choose for this study) to 800 in its simpler configurations (Gutmann et al., 2016). However, the linear theory simplification presents some

limitations when predicting the motion of the atmosphere, such as interactions between waves and turbulence (Nappo, 2012) or the lack of explicit convection. Despite these limitations, ICAR has been shown to be a valuable tool for downscalling proposes showing a good performance compared with observations (Horak et al., 2019), as well as compared with fully dynamical WRF simulations

(Gutmann et al., 2016). Figure 2 shows how the ICAR model was able to improve the 2 m air temperature data, compared with ERA5 reanalysis. This effect is caused by the coarser ERA5 resolution, that smooths the terrain causing warm biases. This is particularly evident in the Lebanon ranges were the elevation gradient ranges from 0 to 3000 m a.s.l. in approximately 25 km (Figure 1). Despite the obvious

improvement in the temperature performance, the simulation is biased towards slightly higher temperatures than in the AWS data. However, the main temporal patterns and the magnitude of the temperature are well represented.



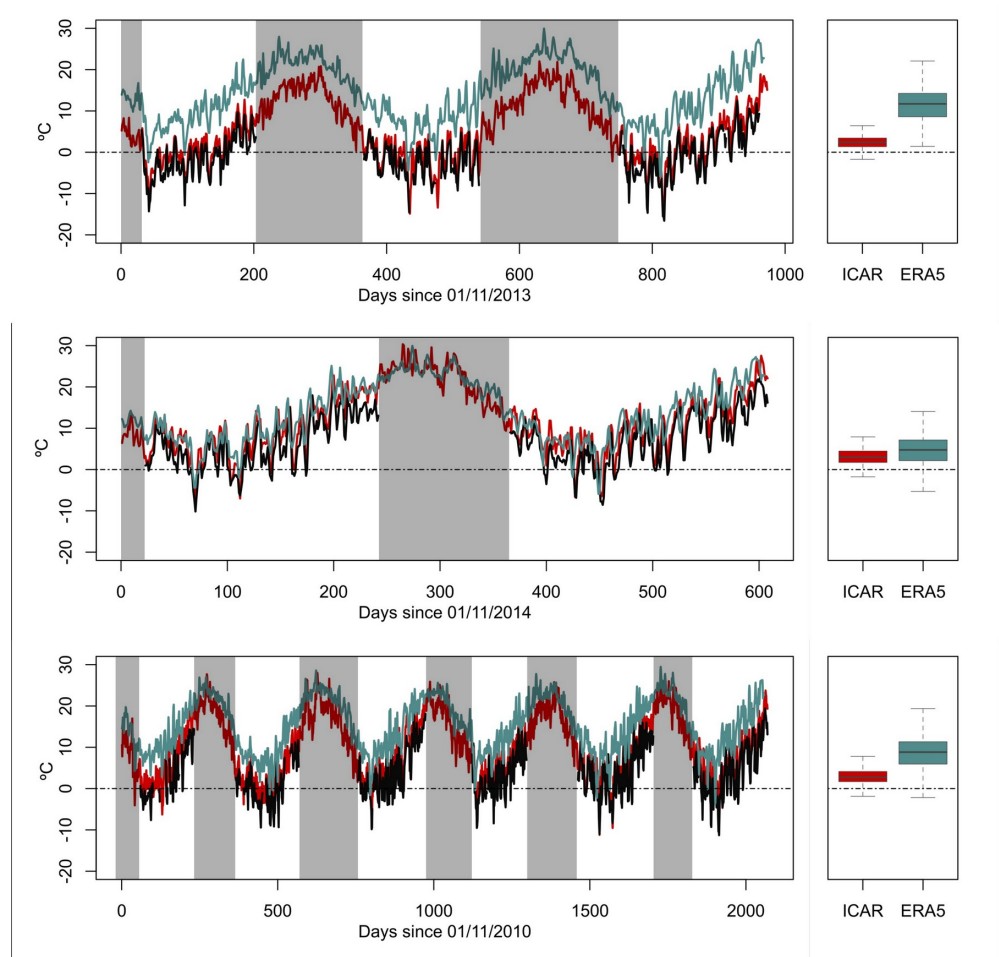

*Figure 2: ERA5 (blue), ICAR (red) and automatic weather stations (black) weekly temperature data. The boxplots represent the distribution of the errors and the gray shadows the data gaps in the observations.*

Similarly, precipitation outputs of ICAR were compared with the gauges deployed in
two of the AWS sites. ICAR reduces the spread of the daily precipitation errors of
ERA5 as shown in Figure 3, even though the ERA5 error are already surprisingly
low considering the spatial resolution and the fact that precipitation is challenging
to simulate by numerical models especially over complex terrain (Legates, 2014).
This validation provides a range of uncertainty estimates to generate the probability
density functions for the perturbations of the ensemble. The selected parameters to
define the shape of the normal probability density function which defines the



additive perturbation index of the temperature were set to a mean of -3.0 ºC and a variance of 1.8 ºC. Similarly, the parameters of the lognormal probability density function used to obtain the multiplicative perturbation factors for the precipitation

were a mean of 2.0 and a variance of 0.75. Even though the parameters were designed to model the uncertainty of ICAR, they are similar to comparable implementations of the PBS (Cortés et al., 2016). Through the forced increase of the variance of the probability density functions, we ensure that the ensemble of snow simulations covers the expected uncertainty space of ICAR, while the PBS has

proved to be robust to progressive variations of the perturbation parameters (Cortés et al., 2016).

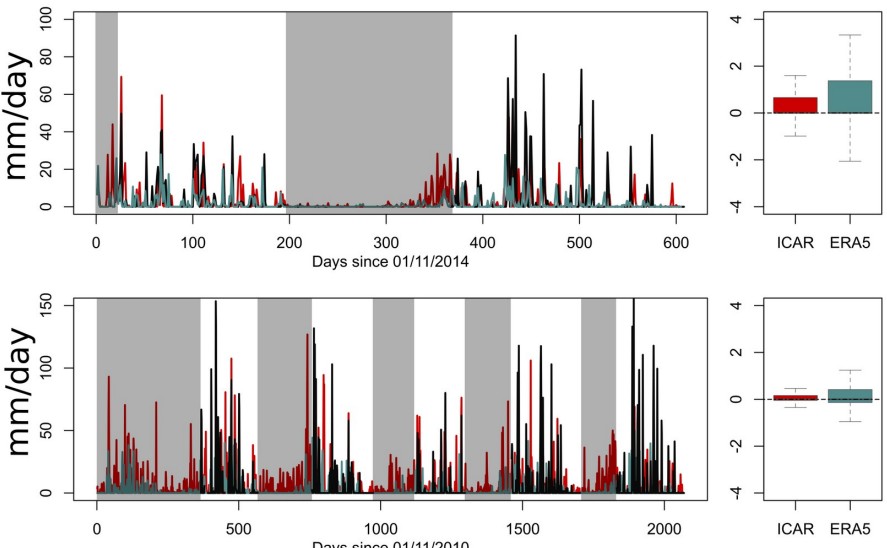

Figure 3: ERA5 (blue), ICAR (red) and automatic weather stations (black) daily precipitation data. The boxplots represent the distribution of the errors and the gray shadows the data gaps in the observations.

## 4.2 Fractional snow cover assimilation

The performance of ICAR_assim was compared against snow depth measurements

at the AWS locations (Figure 4) and MODIS gapfilled products (Figures 5 and 6). In general, ICAR has a tendency to underestimate the SWE compared with ICAR_assim.




This is likely related to the warm biases detected in the simulation, combined with the limitations of the snow model implemented in the Noah land surface model used by ICAR (Barlage et al., 2010). Thus, future versions of ICAR with better

representations of the snow processes through the use of more complex land surface parametrizations like Noah-MP (Niu et al., 2011), as used in the parent WRF simulation, could potentially improve the accuracy of ICAR's SWE outputs (Suzuki and Zupanski, 2018). This effect could be particularly enhanced in the mild climatic conditions of Lebanon, as larger disagreements in the SWE outputs between Noah

and Noah-MP under warm conditions (Kuribayashi et al., 2013). However, the improvement of the snow representations of ICAR is obvious compared with ERA5 reanalysis as it was not able to reproduce the snowpack at all as a result of its coarse resolution.

The results of the validation of ICAR_assim show a good agreement with the

observations. For the estimated SWE, the root mean squared error (RMSE) and the mean absolute error (MAE) relative to the AWS were 189.2mm and 104.52mm respectively after removing the summer from the analyses, with a coefficient of correlation (R) of 0.75 for the annual mean SWE accumulation. Even though ICAR_assim generally shows a good agreement with the observations (especially

considering the scale mismatch between the stations and ICAR_assim), some clear differences were found. Figure 4 exhibits a surprisingly high difference in the magnitude of the observed SWE and the ICAR_assim output for the 2011/2012 season in the third AWS. However, independent observations in the area have described an exceptional snowpack during the 2011/2012 season with snow depths

more than 6 m even reaching up to 10 m locally (Koeniger et al., 2017). Such disagreements between the AWS information and the independent observations can be explained by the high spatial heterogeneity of the snow depth at point scales (López-Moreno et al., 2011). This effect was studied in depth in the Atlas mountains, where the agreement of the snow simulations rapidly drops using resolutions over

250 m (Baba et al., 2019). Such spatial heterogeneity has been shown to be particularly high over mount Lebanon due to the important role of the wind redistribution as consequence of geomorphology (Fayad and Gascoin, 2020). For example, Fayad and Gascoin (2020), reported large differences with the AWS data from insitu measurements on 15 of January 2016, when they measured snow depths

up to 258 cm on the surroundings of the third AWS location (Figure 4; bottom panel), while the AWS sensor itself detected 7.5 cm. However, the comparison between the temporal patterns of the snow cover over Lebanon from MODIS gap-filled daily products and ICAR_assim have shown good levels of agreement with a RMSE=270.2 km$^2$, a MAE=124.1 km$^2$ over a total surface of 4412km$^2$ (Figure 5), and

a Pearson correlation value of R=0.88 in the annual maximum of the snow cover extent (Figure 5). The larger spatial support of the MODIS products permits a more representative and extensive validation of ICAR_assim. Thus, the good agreement between both snow cover products and the generally comparable SWE magnitudes





with the AWS observations shows the temporal consistency of the ICAR_assim
reanalysis.

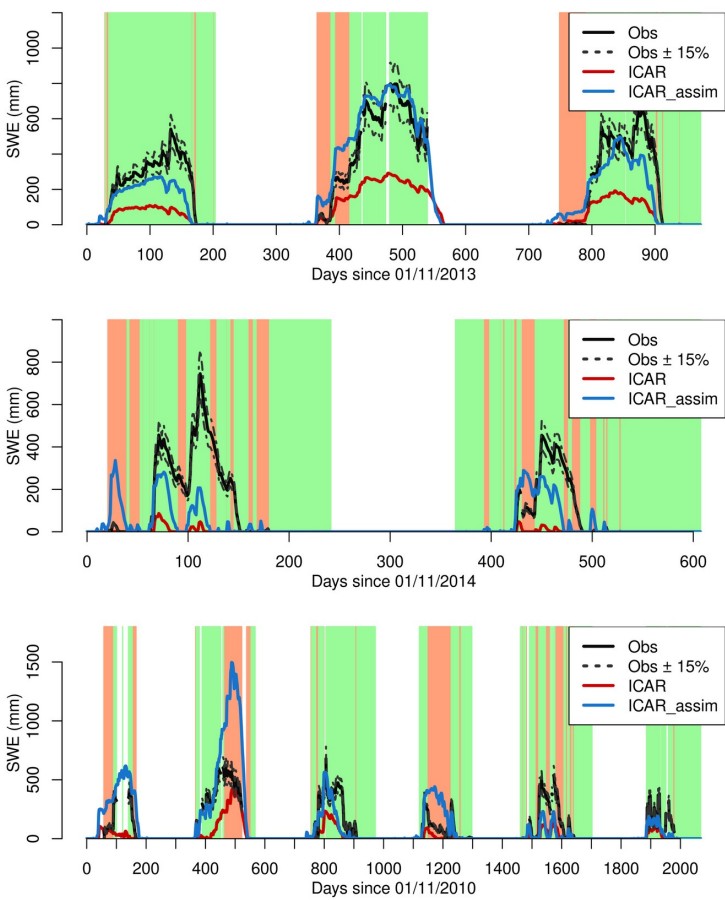

*Figure 4: Comparison between observed, ICAR and ICAR_assim.
SWE products. The green in the background indicates the time
steps when ICAR_assim improves the performance of ICAR.*





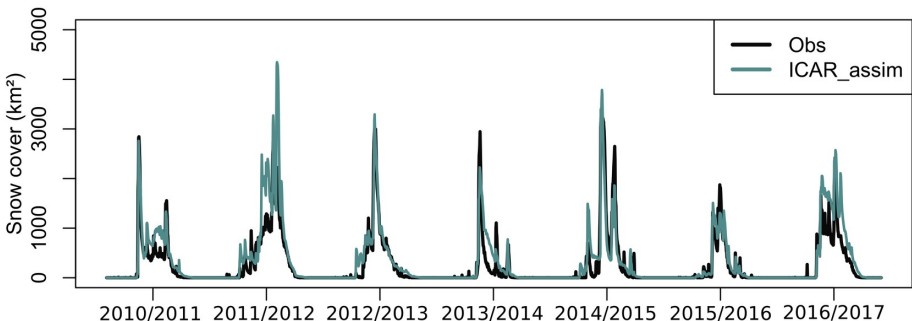

*Figure 5: Daily snow cover extent comparison between MODIS gapfilled products and ICAR_assim.*

The spatial patterns of ICAR_assim, were also compared with the MODIS gapfilled products. The spatial comparison of the $P_{(snow)}$ showed a very good level of agreement with a correlation value of R=0.98, a RMSE=3.0 % and a MAE=2.3 %. There was a general tendency to slightly underestimate the $P_{(snow)}$ values by ICAR_assim specially at the lower elevations. We hypothesize that this effect could

be caused by the selection of a constant SWE depth to calculate the snow cover from the ICAR_assim product. Thus, the shallow snowpacks whose SWE values are under the selected threshold are not recorded as snow presence in the ICAR_assim even though they could potentially be detected as snow by the MODIS sensor. In addition, the MODIS snow cover products should be considered less accurate over areas of

fast melting (Gascoin et al., 2015). In summary, our results have shown how ICAR_assim can accurately reproduce the interannual and intrannual spatiotemporal patterns of the snow cover, with a SWE magnitude comparable with independent observations that agree in its temporal patterns.

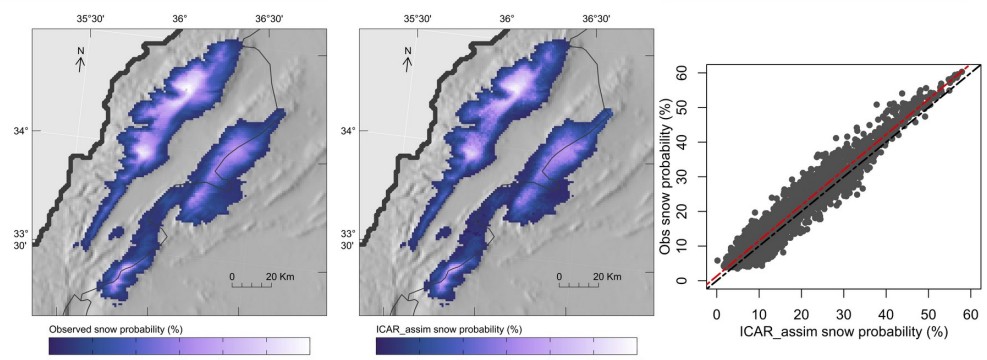

*Figure 6: Snow probability spatial comparison between observed MODIS products and ICAR_assim.*

## 4.3 Snowpack dynamics over Lebanon mountains

Figure 7 shows the spatial distribution of the mean peak SWE values and its temporal coefficient of variation for the recent years. Such values can be influenced by the fact that the study period is relatively humid compared with the previous years (Cook et al., 2016), showing slightly higher values than a long term climatology. However, the length of the reanalyses constitutes a representative sample of the main snowpack dynamics over the region . The snowpack over Lebanon has exhibited the high temporal variability that is characteristic of the Mediterranean snowpacks (Fayad et al., 2017b), with similar values of the coefficient of variation as those observed on other Mediterranean mountain ranges (Alonso-González et al., 2020). The maximum accumulations reach 2000 mm of SWE and are located at the higher elevations of mount Lebanon, where there is a plateau over the elevation of the winter zero isotherm (Fayad and Gascoin, 2020). The temporal coefficient of variation of the annual peak SWE follows unequal spatial patterns, tending to exhibit higher values over the areas sheltered from direct intereaction with the warm and moist Mediterranean air, in addition to a decreasing trend with elevation (Figure 8) as found in other Mediterranean ranges (Alonso-González et al., 2020), reaching a mimimum of 15%.

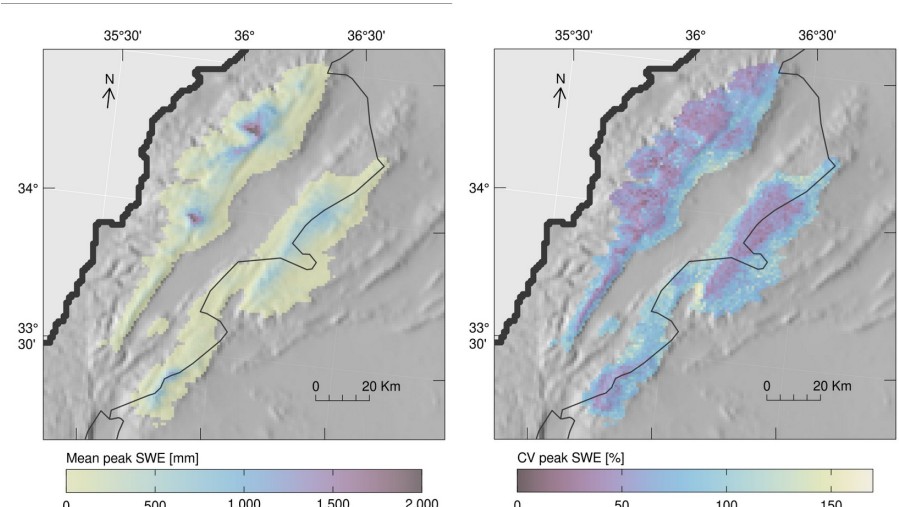

*Figure 7: Averaged annual peak SWE (left) and annual coefficient of variation (right).*

There are obvious differences between the Lebanon and Anti-Lebanon ranges, that can be just partially explained by their different orography. Despite the closeness of both Lebanon and Anti-Lebanon ranges, they exhibit different relationships between the values of P(snow) (Figure 8 right panel) and mean peak SWE (Figure 8 left panel) with the elevation, showing that the differences are not just related to the

particular orography of each range, but also with its climatological characteristics. Thus, at comparable elevations mount Lebanon tends to show higher values of P(snow) and mean peak SWE, with lower values of coefficient of variation, suggesting thicker, longer lasting and seasonally ensured snowpack. The orographic precipitation caused by the uplift of the Mediterranean moisture is a major source of

precipitation in the area (Jomaa et al., 2019), that is probably why Anti-Lebanon mountains shows lower peak accumulations than Mount Lebanon, with Anti-lebanon in the rain shadow leading to lower precipitation and snow accumulation. However, despite the differences in the coefficient of variation values, they tend to become similar at the higher elevations. The same coefficient of

variation occurs in the elevations where the precipitation leads the snow accumulation while they differ at the lower elevations, where the accumulation is conditioned by the temperature. This effect suggest warmer conditions on the Anti-Lebanon mountain as consequence of leeside wind effects (Foëhn type effect), and confirm the sensitivity of the snow simulation to the chosen partition phase method

over Mediterranean mountains (Fayad and Gascoin, 2020).



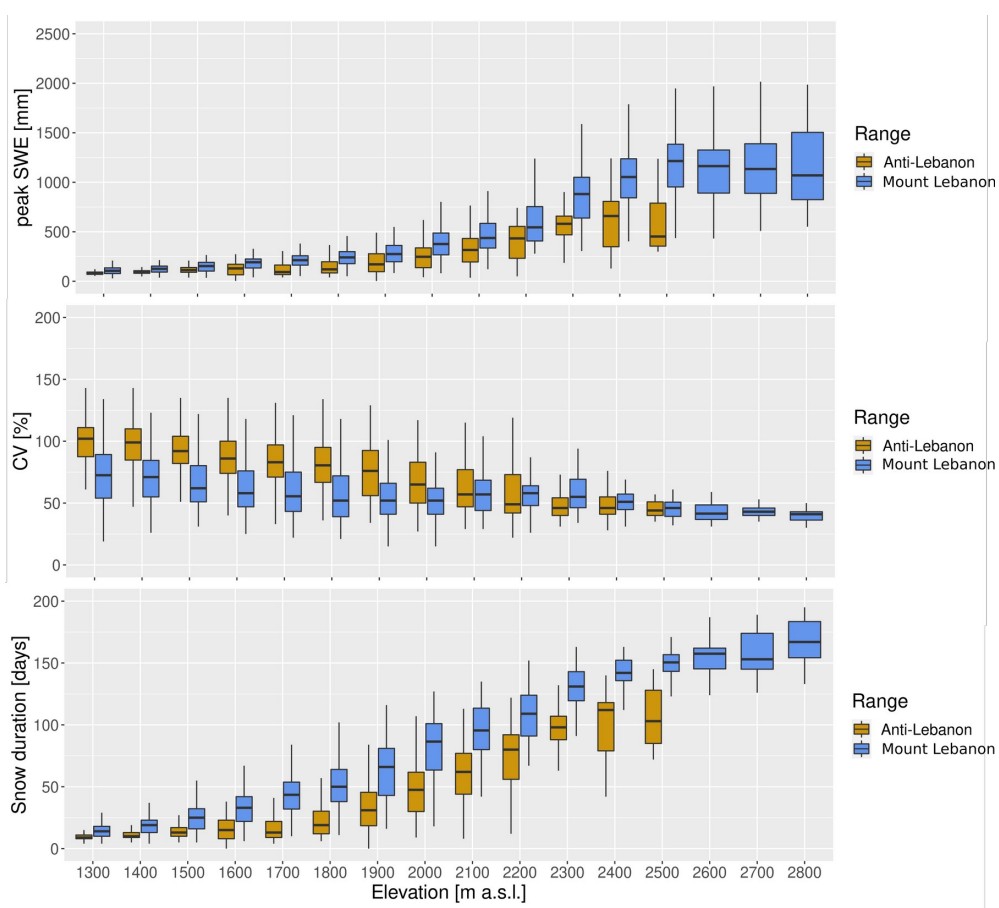

*Figure 8: Relationship between annual peak SWE and elevation (top), coefficient of variation and elevation (middle), and snow duration and elevation (bottom).*

Figure 9 shows the averaged seasonal SWE accumulation at different elevations over both the Lebanon and Anti-Lebanon ranges. Each elevation represents the aggregated pixels of the elevation with a range of ± 50 m a.s.l. For reference, they show on average a peak SWE of 306 mm at the elevation band of 2000 m a.s.l., which is comparable to those found in the Iberian Peninsula mountain ranges (Alonso-González et al., 2020). More specifically, the peak SWE and duration values shows intermediate values between the Central Iberian and Pyrenees ranges at 2000 m a.s.l, but with a peak SWE coefficient of variation of 53 %, that is greater than the highest values of Iberia located at Sierra Nevada with 34 %. The relative area lying at each elevation compared with the total elevation over 1300 m a.s.l. is represented to highlight the importance of the hypsography from the hydrological point of view.





Thus, Lebanon exhibits a deep and long lasting snowpack with up to 1000 mm of
peak SWE on average particularly over 2500 m a.s.l., but the relative areal coverage
of such elevations is very low. This suggest that the mean peak SWE series at lower
elevations could hide a large variation in mass due to the wider areas at lower
elevations, as Alonso-González et al.(2020) found in the Iberian mountain ranges.

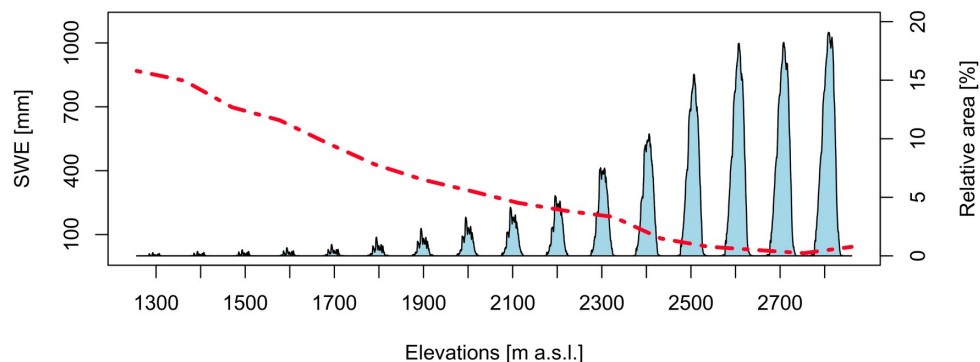

*Figure 9: Mean annual evolution of SWE at different elevation bands (blue) and the
relative areal coverageof each elevation above 1300 m a.s.l (red).*

The thick snowpacks found at the higher elevations are not necessarily the biggest
fresh water resources available due to the hypsometry of the mountain area. Figure
10 shows about the average amount of freshwater stored in the snowpack per
elevations band. It is obvious that the maximum amount of freshwater is stored
between 2100 to 2500 m.a.s.l., despite the fact that thicker snowpacks are at higher
elevations. The cumulative water storage in the snowpack is more than double in
the medium elevation zone (average maximum up to 520 $Hm^3$ from 1300 to 2300m
a.s.l.) when compared to the higher areas (average maximum up to 201 $Hm^3$ at 2400
m a.s.l. and onward), been an important part of the yearly water budget, as mean
annual precipitation was estimated in 7200 $Hm^3$ for the period (2010-2016) (Jaafar
et al., 2020). Noting that this in contrast to the fact that the orography of Lebanon
encourages the storage of snow in the upper areas because of the existence of a high
elevation plateau (Fayad et al., 2017a; Fayad and Gascoin, 2020). This result
suggests new challenges on the water management of Lebanon in the future as a
consequence of climate warming. The snowpack at low elevation areas is more
sensitive to warming (Fayad et al., 2017a; Fayad and Gascoin, 2020). These results
suggest new challenges for the water management of Lebanon in the future as a





consequence of warming climate. The snowpack at low elevation areas is more sensitive to warming (Jefferson, 2011; Marty et al., 2017; Sproles et al., 2013), particularly over areas with mild winter conditions as has been shown in other
Mediterranean regions (Alonso-González et al., 2020).

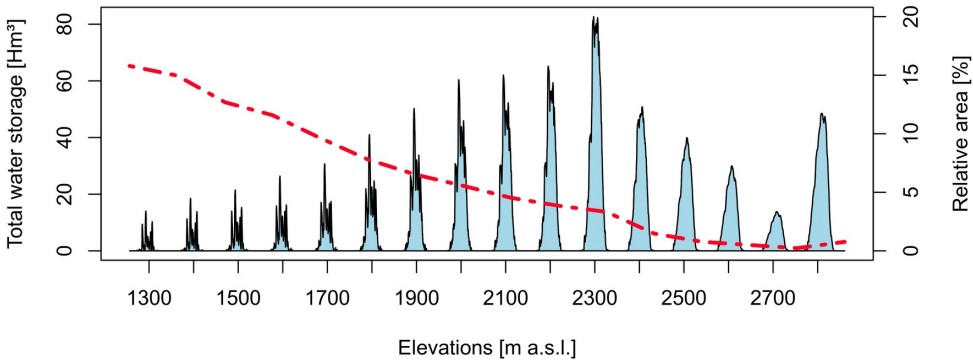

*Figure 10: Averaged annual water stored in the snowpack per elevations.*

## 4. Conclusions

The assimilation of MODIS fSCA through the use of the PBS has proven to be a cost effective way to use remote sensing data in snow simulations, and is particularly
appropriate for simulating snow in data scarce regions. Thus, the generated SWE products show good agreement with MODIS snow cover gapfilled data, with R = 0.98, RMSE = 3.0 % and MAE = 2.3 % for the spatial map of the probability of snow. The time series of snow cover showed a R = 0.88, RMSE = 270.2 km$^2$, and MAE=124.1 km$^2$ over a total surface of 4412km$^2$. The performances in terms of SWE
magnitude with the few available point-scale observations with R = 0.75, RMSE =189.2 mm, and MAE = 104.5 mm after removing the summer from the analyses.

The snowpack over Lebanon is characterized by a high temporal variability. Some differences exist between its two main mountain ranges. Thus, Mount Lebanon exhibits thicker, longer and more regular snowpacks compared to the Anti-Lebanon
range. Such differences cannot only be explained by the elevation difference but also reflects the dryer conditions on the leeside of the Mount Lebanon range due the rain shadow effect. The hypsometry of Lebanon results in the most important snow freshwater reservoir being in the middle elevations (2200-2500 m a.s.l.). Snowpacks at these elevations close to the 0ºC isotherm are highly vulnerable to climate



warming. As such, our findings suggest big challenges for the future management of water resources over the Lebanon region.

**Acknowledgments:** Esteban Alonso-González is supported by the Spanish Ministry of Economy and Competitiveness (BES-2015-071466). This study was funded by the Spanish Ministryof Economy and Competitiveness projects CGL2014-52599-P10

(IBERNIEVE) and CGL2017-82216-R (HIDROIBERNIEVE). We acknowledge support of the publication fee by the CSIC Open Access Publication Support Initiative through its Unit of Information Resources for Research (URICI).

**Code and data availability:** WRF code can be downloaded from https://www2.mmm.ucar.edu/wrf/users/downloads.html. ICAR code can be found

at https://github.com/NCAR/icar. FSM2 is archived at https://github.com/RichardEssery/FSM2. The meteorological data can be found at https://doi.org/10.5281/zenodo.583733. Due to its size, the simulations will be shared under request.

**Author Contribution:** EAG: Conceptualization, Methodology, Writing – original

draft, Software, Data curation, Validation, Visualization. EG: Methodology, Software, Supervision, Writing – review & editing. KA: Conceptualization, Methodology, Software, Writing – review & editing. AF: Methodology, Conceptualization, Writing – review & editing. SG: Conceptualization, Data curation, Methodology, Supervision, Writing – review & editing.

**Conflicts of Interest:** The authors declare no conflict of interest in this article

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
