# Peer review of "Snowpack dynamics in the Lebanese mountains from quasi-dynamically downscaled ERA5 reanalysis updated by assimilating remotely-sensed fractional snow-covered area"

_Hydrology and Earth System Sciences, 2020_

## Short Comment (SC1) · 15 Jul 2020

Dear authors,

thank you for the interesting study and application of ICAR!

In our own study (Horak et al. 2019) that is referenced in your work as well we noticed a strong dependence of the amount and pattern of precipitation on the chosen model

top elevation. To quote the relevant section in the discussion:

"The sensitivity studies leading to the choice of the model top at 4 km have shown that the model top elevation greatly influences precipitation amounts and, in turn, the mean squared errors obtained, see Fig. 2. It is not immediately obvious though why precipitation amounts decrease (not shown) and the MSEs deteriorates for higher model tops. Potential reasons are the influences of divergences in the forcing wind field on the ICAR wind field or numerical artifacts arising from the treatment of the model top in ICAR."

Did you examine your results with regards to their sensitivity to the model top elevation or was there no such dependence? While, with regards to the methodology employed in Horak et al. 2019, a model top set at 4.1 km above topography was seemingly best suited for the South Island of New Zealand, I would anticipate different results for other domains.

Thank you and with kind regards!

Reference: Horak, J., Hofer, M., Maussion, F., Gutmann, E., Gohm, A., and Rotach, M. W.: Assessing the added value of the Intermediate Complexity Atmospheric Research (ICAR) model for precipitation in complex topography, Hydrol. Earth Syst. Sci., 23, 2715–2734, https://doi.org/10.5194/hess-23-2715-2019, 2019.

---

## Short Comment (SC2) · 28 Jul 2020

Dear authors,

Thank you for this valuable contribution. I appreciate that you combine innovative downscaling, remote sensing and data assimilation techniques to improve our knowledge on snowpack dynamics in a high-impact area. I have a small comment on a quote of one of our papers which I found slightly misleading. Lines 100-106:

"However, less often, numerical modeling and remote sensing have been combined in a data assimilation framework to study the multiyear snowpack dynamics. Assimilation of remoted sensed snow cover observations has been shown to improve numerical snowpack models outputs in both distributed (e.g. Baba et al., 2018; Margulis et al., 2016) and semi distributed simulations (Cluzet et al., 2020; Fiddes et al., 2019). These approaches are particularly promising in data-scarce regions to reduce the biases in atmospheric forcing."

I'm afraid that the reader might understand that in Cluzet et al., 2020, we assimilated data in a semi-distributed setting, while we were indeed unable to assimilate any data due to significant biases in the satellite product retrievals. In this study, we only suggest that assimilating satellite reflectances could be beneficial in a semi-distributed area provided that this bias issue is fixed (Lamare et al., 2020), by exhibiting strong correlations between observed and modeled variables in a wide diversity of topographic conditions. I think that Fiddes et al., 2019 go much more forward in demonstrating the performance of data assimilation in such a framework, although with a different variable. More recently, we submitted a paper in GMD (Cluzet et al., 2020b) where we actually assimilate reflectances (and snow depth) in a semi-distributed area, but assimilating only synthetic data (i.e. model outputs), showing some potential, but still being unable to assimilate real data. This citation might be more appropriate depending on your purpose.

So to wrap up, I would suggest to either unquote Cluzet et al., 2020, or reformulate the sentence to make it clear that we did not assimilate real data, or cite Cluzet et al., 2020b underlining that it's only a theoretical experiment.

Thanks a lot again for this manuscript, and best regards !

References: 1. Cluzet, B., Revuelto, J., Lafaysse, M., Tuzet, F., Cosme, E., Picard, G., Arnaud, L. & Dumont, M. (2020). Towards the assimilation of satellite reflectance into semi-distributed ensemble snowpack simulations. Cold Regions Science and Technol-

ogy, 170, 102918. 2. Cluzet, B., Lafaysse, M., Cosme, E., Albergel, C., Meunier, L. F., & Dumont, M. (2020). CrocO_v1. 0: a Particle Filter to assimilate snowpack observations in a spatialised framework. Geoscientific Model Development Discussions, 1-36. 3. Lamare, M., Dumont, M., Picard, G., Larue, F., Tuzet, F., Delcourt, C., & Arnaud, L. (2020). Simulating Optical Top-Of-Atmosphere Radiance Satellite Images over Snow-Covered Rugged Terrain. The Cryosphere Discussions, 1-46.

---

## Referee Comment (RC1) · Anonymous Referee #1 · 8 Aug 2020

The authors simulated the snowpack, more specifically developed a 1 km regional scale snow reanalysis (ICAR_assim) covering the period 2010-2017, for the Lebanon Mountains, as an alternative to sparse snowpack observations. The authors made use of different data sources of remote sensing and atmospheric reanalysis data, moreover they combined different techniques as downscaling, data assimilation, energy and mass balance modeling. In these respects, the work is a very valuable and excellent contribution to the science especially for the snow studies in mountainous re-

gions where monitoring is always difficult and observation network is generally sparse despite the importance of snow in water resources.

The couple of questions are listed below to improve the evaluation of results and make it more explanatory for the community working on snow studies:

1) It is indicated in the article that "ICAR model was used to obtain a finer 1 km x 1 km spatial grid atmospheric simulation nested in the aforementioned WRF simulation domain.", and the comparisons and error analysis are done for ICAR and ERA5 for temperature and precipitation (Figure 2 and 3), showing that ICAR better performs compared to ERA5, which is rather expected concerning the scaling and processes. It would also be important to see the performance of WRF in comparison with ICAR so that one can be sure that ICAR is superior to WRF and it is worth to do such a downscaling process even though it is rather preferable compared to fine scale WRF simulation. This is also valid for the comments on reproduction of snowpack by ICAR and ERA5.

2) It is better to include topographical and climatological characteristics of AWS (e.g. altitude, aspect, annual average values etc) and comment on these since there are differences in comparison results (e.g. the error difference is less (Figure 2) in the second AWS, it could be assigned to the topographic similarity or just the short period of comparison, but the errors are rather high for the same station in precipitation comparison). This would also be helpful for SWE comparisons in Figure 4.

3) The comparison in observed and simulated SWE values is very valuable and worth further discussion. The authors give some details on the inconsistency of comparisons in the third AWS for 2011/2012 which indicates that the observed SWE values might have rather higher values. On the other hand, this inconsistency is also valid for independent snow cover comparison in Figure 5 for the same year, which might indicate some other problems for that year. The consistencies are rather high for the first AWS may be due to its higher altitude, however especially for the second AWS, neither ICAR nor ICAR_assim provides a good performance, for the third AWS, there are varying comparison results and the scale of SWE (due to extreme value in 2011/2012) makes the graphic rather difficult to interpret. Questions arise on the differences in ICAR and ICAR_assim; assimilation process changes ICAR results dramatically in some years (the second AWS, both years but especially 2015/2016; the third AWS, 2010/2011, 2013/2014) while not much for the other years. In some years, assimilation yields significant amount of SWE from almost no snow condition (e.g, the third AWS, first year). On the other hand, assimilation shows very well performance on the first AWS for 2014/2015. Would it be possible to give some explanations on such a big and varying impact of assimilation?

4) Concerning a rather constant (or slightly decreasing) relative area in Figure 9 and rather constant SWE values above 2500 m a.s.l. in Figure 8, it is surprising to see an increase in total water storage at 2800 m a.s.l. so it would be nice to give attention to this part.

Some technical correction comments are:

1) Since ICAR_assim is already produced by assimilating MODIS through ICAR, the comparison in Figure 5 might include ICAR directly instead of ICAR_assim and/or more statistical results can be given on both.

2) In section 4.3, the time period (2010-2017?) should be indicated instead of "recent years" for the explanation of Figure 7.

3) In paragraph with code "515" there is a repetition for two sentences which should be avoided.

---

## Referee Comment (RC2) · Anonymous Referee #2 · 9 Aug 2020

The authors simulated the snowpack dynamics over the Lebanon mountains, combining several data sources (reanalysis data, remote sensing data, in situ observations) and techniques (regional atmospheric modelling, snow modelling, data assimilation techniques). The study is a useful contribution for understanding the role of snow in a mountainous area in a semiarid region, with limited information and where water resources are highly dependent of these dynamics. Despite the topic is of interest to the scientific community, there are certain issues regarding methodology, result assess-

ment that might be better addressed.

General Comments

Throughout the paper many assumptions regarding the selected methods are done. In most of the cases a clear justification of why these are the options selected are missing. Moreover, due to the numerous methodologies of different natures used, sometimes the methodology section become difficult to follow (see specific comments below). Authors should have taken more advantage of the results, with deeper assessment and discussion. There are also aspects to improve in the visualization of the results.

Specific comments

Line 45: Could the authors be more precise with deep and long lasting snowpacks? How deep and how long? Are those common characteristics of snowpack in all Mediterranean mountains?

Line 47 and 48 and Line 51: Does the interannual variability referred in Line 51 any influence in the reshape of the hydrographs? I might think yes.

Lines 100-102: There are many works in combining numerical modelling and remote sensing using data assimilation techniques. Please try to be less categorical in your statement.

Lines 107-126: I miss the aim of the research in this paragraph, what is relevant scientific question addressed by this work? The paragraph seems more a summary of a methodological section.

Line 129: Could the authors provide some data or references about this typical Mediterranean climatology?

Line 139: The authors mentioned here that the mountainous ranges act as a barrier to humidity advected from the sea. Could you provide some data about physiographic (elevation, slopes, land covers) and meteorological characteristics that differentiate both

mountainous ranges?

Line 158: Could the authors clarify what are they referring with previously?

Line 162: Why do the authors chose 35 levels and 50hPa?

Line 166: Which version of WRF is been used?

Line 172-180: The authors justify the use of a specific parameterization schemes in the WRF simulation base on Ikeda et al., (2010) a study performed over Colorado. I can understand similitudes regarding topography of both areas. However, the sizes of both mountainous ranges, the proximity to the sea are different. What are the influences of having choose the same parameterization? Since the lack of data does not allow a deeper analysis, could you explain a bit deeper the physical reasons behind the selection of this atmospheric parameterization?

Line 204: Do the authors mean they are not considering any convective process in their simulation? What are the implications?

Line 220: What is the temporal resolution of Theia? And therefore, how many days of overlapping between Theia and MODIS do you have? Is this overlapping constant during the year? Could that introduce errors in the transformation function?

Line 226-228: How do the authors choose this 40% of the data? Why do the authors use a bigger number of data for calibration than for validation? Could the authors show the same errors in the calibration phase to see differences? I think the reader would be interested in see the fitting graphs.

Line 234: The authors mentioned here the difference between revisiting times of Aqua and Sentinel-2, but what is the Terra revisiting time in the area?

Line 240: When the authors say "empty" are they referring to a non-snow cell or a non-information cell? Why have the authors chosen that option instead of an interpolation with nearest cells?

[Figure]

Line 259: How do the authors apply the FSM2 snow model in a distributed way? In the last paragraph of this section it seems the authors use some depletion curve for that. However, it is not clear if that is just part of the assimilation or plays a role in the actual snow modelling. Could the authors add a sentence in this paragraph specifying how that is done?

Line 270: Why do the authors chose a log-normal and a normal gaussian probability density function? Are just precipitation and temperature the inputs/forcing variables of the FSM2 snow model? If they are more than precipitation and temperature, how are you perturbing them in the assimilation scheme?

Line 319: How was the snow depth measured? Why not to do the comparison in term of snow depths avoiding to use a constant density value? Reading Essery (2015) FMS2 provides snow depth as an output.

Lines 328-342: The methodology explained here is not clear. Neither the reasoning behind nor the way SWE is compare against satellite observation. Are you using SWE measuring using remote sensing?

Lines 345-381: I miss numbers supporting the statements throughout the section. Here a few examples: "Figure 2 shows how the ICAR model was able to improve the 2 m air temperature data, compared with ERA5 reanalysis", "ICAR reduces the spread of the daily precipitation errors". Moreover, I think it could be interesting to analyse a deeper when the error between observations and simulations occurs. Are they bigger in winter than in summer? Is there any dependence with the total precipitation of the hydrological year (dry or wet)? It may have large impact in your results.

Figure 3: In general, ICAR precipitation values seems to be higher than ERA5 precipitation values. However, the bias in ERA5 are positive and bigger than ICAR. How do the authors explain that?

Figure 4: How do the authors explain the heterogenous differences in the assimilation

results between years? The authors gave some explanation about one of the years in Lines 399-410, however, could you give deeper explanations about the differences between years in the whole period?

Lines 421-448 and Figure 5: If I read well, here you are comparing the results of your assimilation (ICAR_assim) with the assimilated variable (Obs). This is a prove that your assimilation scheme works well, and therefore, the obtained metrics should be interpreted as that. The real impact of the assimilation scheme on snow dynamics is the show in the comparison with the independent variable SWE, not assimilated during the process.

Section 4.3: All section is written as if the simulated values were a "ground truth", I would indicate some of the limitations of the performed simulaitons and all the sources of uncertainty and errors that are conditioning these statements.

Figures 9 and 10: What does the relative area referring to (snow area over the area of the band or area of the band over all area of the mountainous ranges)? It would be interesting to see these two graphs in both mountainous ranges.

Lines 504-508: Could you elaborate more the reasoning in this paragraph?

Figure 10: How do you explain that the total storage at 2800 m a.s.l. increases?

Technical comments

Figure 1. What are A and B, could the authors specify it in the body text lines 58-62 and in the figure caption.

A Figure with a scheme of the implementation process would help to better understand the complexity of the flow chart followed.

Figures 2, 3 and 4: It is difficult to know in which season of the year you are with the format "Days since". I propose to add actual dates in x-axis of these figures. Moreover, it is complicate to see differences between the 3 represented variables, especially in

the precipitation graph. Finally, it is difficult to see what the values of the boxplot are represented, I would recommend here to change the y-axis limits, add, y-axis values and/or a grid.

Line 524: A sentence is repeated.

---

## Referee Comment (RC3) · Anonymous Referee #3 · 29 Aug 2020

The paper presents an approach to downscale ERA5 reanalysis by using MODIS fSCA information. Even though the approach is not completely innovative,the research has a high relevance for the application in arid areas. Below detailed comments. p.3 line 107: please provide here a clear statement about the objectives of the work and the innovative part with respect to the current literature. Section 3..2.1. More detailed information about the processing of MODIS data need to be introduced here. please add the new adapted linear function that the authors found by using Theia data and the explanation

why it differs from the Salomonson&Appel2004. How were MOD and MYD images merged? is there a cloud gap filling procedure? If the use of MYD produces a higher error why do not use only MOD? was a validation with ground measurements conducted? this can provide an independent source of information to better quantify the accuracy of the new proposed linear relationship. Moreover in the validation, a comparison of the new linear relationship with the one proposed by Salomonson&Appel2004 is advisable to understand the advantage of the new approach. p.9, line 320: as the density can change during the season, please justify the use of the value.

---

## Author Comment (AC1) · 10 Oct 2020

Dear Johannes Horak, Thank you very much for your comments. As you noticed, the performance of ICAR is very sensitive to the setup of the model top. As a very novel model, more research should be developed related to the dependence of ICAR's performance on the choice of model top, including the physical and numerical reasons involved in such a dependence as it can greatly increase its applicability over ungauged

regions. Unfortunately, there is not enough observational data over Lebanon to develop a sensitivity study for such variable, a limitation faced in choosing all parameterizations of the models involved in the study (ICAR, WRF and FSM). Thus, we had to choose the most likely configurations of the models considering the existing references in the topic, including Horak et al. 2019. However, in our case the uncertainty introduced by the forcing data should be greatly reduced by the fractional snow cover assimilation scheme implemented in the study.

---

## Author Comment (AC2) · 10 Oct 2020

Dear Bertrand Cluzet, Thank you very much for your clarification. We will change the Cluzet et al., 2020 citation to Cluzet et al., 2020b and remark the theoretical nature of the experiment as you suggest. All the best.

[Figure]

335, 2020.

---

## Author Comment (AC3) · 10 Oct 2020

Authors: We would like to thank the anonymous referee for his/her interest and the comments on our manuscript. Bellow we provide a point by point answer to the issues raised by referee #1

**Ref.1: It is indicated in the article that "ICAR model was used to obtain a finer 1 km x 1km spatial grid atmospheric simulation nested in the aforementioned WRF simulation domain.", and the comparisons and error analysis are done for ICAR and ERA5 for temperature and precipitation (Figure 2 and 3), showing that ICAR better performs compared to ERA5, which is rather expected concerning the scaling and processes. It would also be important to see the performance of WRF in comparison with ICAR so that one can be sure that ICAR is superior to WRF and it is worth to do such a downscaling process even though it is rather preferable compared to fine scale WRF simulation. This is also valid for the comments on reproduction of snowpack by ICARand ERA5.**

Authors: It is mostly true that there is an expected improvement of the ICAR simulation compared with ERA5 as consequence of its higher resolution. This is obvious concerning the temperature because of the smoothing of the topography caused by the coarser resolution of ERA5 while it is necessary to remark that there is an added value of using ICAR for simulating the precipitation which is a much more uncertain variable. The main objective of ICAR was to provide the forcing for the data assimilation scheme that is described later. An intercomparison between ICAR, ERA5 and WRF is a valuable exercise that should be carried out, but it falls outside of the scope of the current manuscript. In our study the comparison between ICAR and the automatic weather stations is performed mostly to define the parameters of the prior probability distribution functions used to perturb the members of FSM2 ensemble of simulations, as is underlined in the text:
"This validation provides a range of uncertainty estimates to generate the probability density functions for the perturbations of the ensemble"
An intercomparison of models should be developed over much more well instrumented areas. We do not consider it appropriate o develop an in depth intercomparison herein, as the results could be extremely constricted by the very low availability of data. In the case of snow, ERA5 is not able to even simulate any snowpack for our domain (as highlighted in the text) as a consequence of the coarse resolution. WRF simulation is able to simulate a very marginal snowpack, as mount Lebanon is too small to reproduce the snowpack at 10 km spatial resolution. Thus, the comparison between models will not show similar results, but this does not mean that each model is not working as expected.

**Ref.1: It is better to include topographical and climatological characteristics of AWS (e.g. altitude, aspect, annual average values etc) and comment on these since there are differences in comparison results (e.g. the error difference is less (Figure 2) in the second AWS, it could be assigned to the topographic similarity or just the short period of comparison, but the errors are rather high for the same station in precipitation comparison). This would also be helpful for SWE comparisons in Figure 4.**

Authors: We have included a new table summarizing the topographical characteristics of the automatic weather stations and the pixel elevation of ICAR.

**Ref.1: The comparison in observed and simulated SWE values is very valuable and worth further discussion. The authors give some details on the inconsistency of comparisons in the third AWS for 2011/2012 which indicates that the observed SWE values might have rather higher values. On the other hand, this inconsistency is also valid for independent snow cover comparison in Figure 5 for the same year, which might indicate some other problems for that year. The consistencies are rather high for the first AWS may be due to its higher altitude, however especially for the second AWS, neither ICAR nor ICAR_assim provides a good performance, for the third AWS, there are varying comparison results and the scale of SWE**

**(due to extreme value in 2011/2012)makes the graphic rather difficult to interpret. Questions arise on the differences in ICAR and ICAR_assim; assimilation process changes ICAR results dramatically in some years (the second AWS, both years but especially 2015/2016; the third AWS,2010/2011, 2013/2014) while not much for the other years. In some years, assimilation yields significant amount of SWE from almost no snow condition (e.g, the third AWS,first year). On the other hand, assimilation shows very well performance on the first AWS for 2014/2015. Would it be possible to give some explanations on such a big andvarying impact of assimilation?**

Authors: There is a very big difference of scale between the ICAR/ICAR_assim simulations and the point-scale AWS observations. Much of the inconsistencies could be explained by this scale mismatch, as the snowpack varies at much finer resolutions at the local scale of the AWS's as explained in paragraph 4.2 Fractional snow cover assimilation.

We hypothesize that The 2011/2012 inconsistency between ICAR_assim and MODIS gap-filled snow cover extent could be explained by the fact that MODIS gapfilled products will be biased during the snow seasons with persistent cloud covers (as the 2011/2012 season), as the gapfilling algorithm will have just a few observations. Thus, the ICAR_assim snow cover exhibits higher values than MODIS as consequence of the low elevation snowpacks. While MODIS gapfilled products will not be able to detect properly such low elevation and very variable snow covers as a consequence of the cloud cover, the particle batch smoother is able to propagate the few fSCA observations through the whole season. However, it is very surprising that the independent observations of Koeniger et al., 2017 highlight the extraordinary snowpack accumulations of the 2011/2012 snow season (as can be observed in the ICAR_assim reanalyses), while it is not observed in the MODIS gapfilled products. To improve the discussion about this topic, we have added the following sentences to the manuscript.

"In addition, the MODIS snow cover products should be considered less accurate over areas of fast melting (Gascoin et al., 2015). Such effect combined with the fact that 2011/2012 snow season showed persistent cloud covers related with its exceptional snowpack, could explain the biases in the Figure 5 2011/2012 snow season, as the gapfilling algorithm had less information to fill the MODIS snow cover time series, while the PBS had propagated the fSCA information through the whole season from the few available observations."

The inconsistencies observed between the AWS and ICAR_assim, are similar to those found in Fayad and Gascoin (2020) using the MICROMET + SNOWMODEL framework. They found that it was not trivial to simulate the snowpack at the AWS locations, even using meteorological observational data from the AWS itself. They hypothesize that the inconsistencies could be related to the partitioning of the precipitation phase, because of the relatively warm conditions close to the 0ºC. In addition, some local effects are probably affecting the AWS data, but unfortunately there is not enough information to study such effects and the inconsistencies should be considered as part of the total uncertainty. Actually, the ICAR_assim and AWS SWE comparison (as for any other grided numerical model) should be taken with care, as the ICAR_assim represents an averaged region (i.e. model grid cell). Thus, the good results showed on the first AWS for 2014/2015 snow season could be completely different if the AWS were at a different location just few meters way, as reported by a manual inspection by Fayad and Gascoin (2020) at 15 of January 2016, cited in the manuscript as follows

*"For example, Fayad and Gascoin (2020), reported large differences with the AWS data from insitu measurements on 15 of January 2016, when they measured snow depths up to 258 cm on the surroundings of the third AWS location (Figure 4; bottom panel), while the AWS sensor itself detected 7.5 cm."*

**Ref1: Concerning a rather constant (or slightly decreasing) relative area in Figure 9 and rather constant SWE values above 2500 m a.s.l. in Figure 8, it is surprising to see an increase in total water storage at 2800 m a.s.l.  so it would be nice to give attention to this part.**

Authors: Such an increment is caused by the accumulated surface over 2800 masl, combined with the very high values of SWE at the higher elevations. Actually, in the figures 9 and 10 the relative area above 2800 masl is slightly higher than the previous elevations. We have modified the figures 9 and 10 to include the label >2800 to clarify this.

**Specific comments:**

**Ref1: Since ICAR_assim is already produced by assimilating MODIS through ICAR, the comparison in Figure 5 might include ICAR directly instead of ICAR_assim and/or more statistical results can be given on both.**

Authors: Figure 5 was designed to show the performance of the PBS, that is why we consider it is better to show ICAR_assim snow cover extent compared with the MODIS gapfilled snow cover extent. To highlight the improvement of the performance after the PBS implementation we have added the statistics of ICAR compared to MODIS gapfilled products.

**Ref1:** *In section 4.3, the time period (2010-2017?) should be indicated instead of "recent years" for the explanation of Figure 7.*

Authors: Change accepted, thanks.

**Ref1: In paragraph with code "515" there is a repetition for two sentences which should be avoided.**

Authors: Corrected, thanks

---

## Author Comment (AC4) · 10 Oct 2020

Authors: We would like to thank the anonymous referee for his/her interest and the comments on our manuscript. Bellow we provide a point by point answer to the issues raised by referee #2

**Ref.2: Line 45: Could the authors be more precise with deep and long lasting snowpacks? How deep and how long? Are those common characteristics of snowpack in all Mediterranean mountains?**

Authors: There is not an obvious answer to these questions. The depth and duration of the snowpack are often related to the elevation. As is stated in the text, the Mediterranean snowpacks are deep and persistent, as long as the range has a sufficient elevation, due to the wintertime distribution of the precipitation of Mediterranean climates. The references supporting such a statement (Alonso-González et al., 2020; Fayad et al., 2017b) are much more extensive with several data at different locations and elevations. We have added to the following to the text:
"...snowpacks accumulating more than 3 meters and lasting more than 5 months at the summit areas"

**Ref.2: Line 47 and 48 and Line 51: Does the interannual variability referred in Line 51 any influence in the reshape of the hydrographs? I might think yes.**

Authors: Yes it does. We have modified the text as follows:
"Mediterranean snowpacks are characterized by a high interannual variability, which affect the amount and seasonality of river flows"

**Ref.2: Lines 100-102: There are many works in combining numerical modelling and remote sensing using data assimilation techniques. Please try to be less categorical in your statement.**

Authors: We have removed the following sentence:
"However, less often, numerical modeling and remote sensing have been combined in a data assimilation framework to study the multiyear snowpack dynamics."

**Ref.2: Lines 107-126: I miss the aim of the research in this paragraph, what is relevant scientific question addressed by this work? The paragraph seems more a summary of a methodological section.**

Authors: We have added the following sentence to the text:
"The objectives here are: i) to explore the potential of a methodology to develop a snowpack reanalysis over data scarce regions and ii) to describe the main snowpack dynamics over the Lebanese mountains being the first use of ICAR for this approach"

**Ref.2: Line 129: Could the authors provide some data or references about this typical Mediter-ranean climatology?**

Authors: We have added the following reference to the text:
*Peel, M. C., Finlayson, B. L., and McMahon, T. A.: Updated world map of the Köppen-Geiger climate classification, Hydrol. Earth Syst. Sci., 11, 1633–1644, https://doi.org/10.5194/hess-11-1633-2007, 2007.*

**Ref.2: Line 139: The authors mentioned here that the mountainous ranges act as a barrier to humidity advected from the sea. Could you provide some data about physiographic (el-evation, slopes, land covers) and meteorological characteristics that differentiate both mountainous ranges?**

Authors: We consider that Fig1 describes elevations and slopes and the Jomaa et al. (2019) support such a statement about the orographic precipitation. We have added the following sentence referring to the land cover:

"Lebanese mountains are highly karstified encouraging the infiltration of rainfall and snowmelt. The land cover is mostly composed of bare rocks and soils with irregularly distributed patches of shrubland, oak and pine forest."

**Ref.2: Line 158: Could the authors clarify what are they referring with previously?**

Authors: We have change "Previously" by "First".

**Ref.2: Line 162: Why do the authors chose 35 levels and 50hPa?**

Authors: This is the regular WRF configuration. As there is limited information in our domain, we had to choose a regular model set up. We have added the following sentence to the text:
"similarly to other studies over Mediterranean climate (Arasa et al (2016))"
Arasa, R. , Porras, I. , Domingo-Dalmau, A. , Picanyol, M. , Codina, B. , González, M. and Piñón, J. (2016) Defining a Standard Methodology to Obtain Optimum WRF Configuration for Operational Forecast: Application over the Port of Huelva (Southern Spain). Atmospheric and Climate Sciences, 6, 329-350. doi: 10.4236/acs.2016.62028.

**Ref.2: Line 166: Which version of WRF is been used?**

Authors: We have added "3.8 version" to the text.

**Ref.2: Line 172-180: The authors justify the use of a specific parameterization schemes in the WRF simulation base on Ikeda et al., (2010) a study performed over Colorado. I can understand similitudes regarding topography of both areas. However, the sizes of both mountainous ranges, the proximity to the sea are different. What are the influences of having choose the same parameterization? Since the lack of data does not allow a deeper analysis, could you explain a bit deeper the physical reasons behind the selection of this atmospheric parameterization?**

Authors: As referee.3 highlights there is not a way to test different WRF parameterizations here. Thus, there is a need for assumptions. However, the reason for using an atmospheric model as forcing is actually the lack of observational data allowing to work over areas and times where it is not possible to find any information. There is probably not any physical reason to choose a specific WRF set up. Most of the studies looking for perfect WRF configurations are factorial experiments over well monitored/instrumented areas, as it is not easy to offer physically based explanations about why a particular parameterization performs better than others.
Despite thefact that all the parameterizations used in the WRF simulation (as well as its alternatives) are physically based, there are many empirical components inside them that are impossible to avoid. Thus, we can not justify choosing a parameterezation over the region different from what the literature recommends, as all the parameterizations concern physics and at some point over empirical approximations. This is where the importance of the data assimilation becomes crucial, correcting the uncertainty caused by parameterizations and observations, exploiting the strengths and weaknesses of both.

**Ref.2: Line 204: Do the authors mean they are not considering any convective process in their simulation? What are the implications?**

Authors: Convection can not be represented by the linear theory simplification and therefore by ICAR. The convective schemes of ICAR are highly experimental and in most cases becomes the model unstable making it crash at 1km resolution. The implications for winter precipitation are probably related to the amount of precipitation, but are much less significant than during the summer season. Such effects should be compensated by the PBS if it has some impact on the snowpack. We have added the following clarification to the text:

"The lack of convection could have some impact on the total amount of precipitation, and therefore on the seasonal snowpack. However, such deviations in the total amount of precipitation are compensated by the PBS."

**Ref.2: Line 220: What is the temporal resolution of Theia?  And therefore, how many days of overlapping between Theia and MODIS do you have?  Is this overlapping constant during the year? Could that introduce errors in the transformation function?**

Authors: The revisit period of Sentinel-2 is at least 5 days since the launch of Sentinel-2B (i.e. after march 2017). It can be even less in areas where successive swaths overlap laterally (every 2 and 3 days). As written in the manuscript we have used a total of 645 Sentinel-2 snow images. This corresponds to all available images from 03 Sep 2017 to 24 Dec 2018 over Lebanon (five Sentinel-2 tiles: T36SYB, T36SYC, T36SYD, T37SBT, T37SBU). For every Sentinel-2 image we can match a MODIS image since there is a MODIS image every day over Lebanon during the same period. However, the number of Sentinel-2/MODIS images is a bit misleading since large parts of a single image can be covered by cloud, or correspond to the sea surface. In addition we only extracted Sentinel-2 pixels where MODIS NDSI is strictly positive (i.e. MODIS snow covered pixels) to establish the relationship between MODIS NDSI and Sentinel-2 fSCA. Therefore we think it is more informative to provide the number of pixels that were actually used to optimize the fSCA function (5.84e4). We will clarify this in the text accordingly.

**Ref.2: Line 226-228: How do the authors choose this 40% of the data? Why do the authors use a bigger number of data for calibration than for validation? Could the authors show the same errors in the calibration phase to see differences? I think the reader would be interested in see the fitting graph**

Authors: In fact we used 40% for calibration (L226), therefore we used a bigger number of data for validation. By using a larger fraction of data for validation we expect to have a more robust estimate of the model accuracy. We will include the graph of the model calibration in supplement of the revised manuscript.

**Ref.2: Line 234: The authors mentioned here the difference between revisiting times of Aqua and Sentinel-2, but what is the Terra revisiting time in the area?**

Thank you for this comment, it is approximately 10:30 A.M. local time dailly, i.e. similar to Sentinel-2. We will add this information in the revised manuscript.

**Ref.2: Line 240: When the authors say "empty" are they referring to a non-snow cell or a non-information cell? Why have the authors chosen that option instead of an interpolation with nearest cells?**

Authors: We consider it a non-information cell (changed in the text for clarification). The reason for this is to not  propagate into the reanalysis information derived from interpolations. As a smoother, the PBS can propagate the information over the whole season (forward and backward in time), with this information being the trajectory of the fSCA the variable that is assimilated. There is not any

added value on including a few more noise cells derived from incomplete observations, especially in a Mediterranean area like Lebanon where persistent cloud cover is not expected.

**Ref.2: Line 259: How do the authors apply the FSM2 snow model in a distributed way? In the last paragraph of this section it seems the authors use some depletion curve for that. However, it is not clear if that is just part of the assimilation or plays a role in the actual snow modelling. Could the authors add a sentence in this paragraph specifying how that is done?**

Authors: There is not any specific FSM set-up to implement it in a distributed way. What we did was implement the PBS grid cell by grid cell, generating the distributed reanalysis. The subgrid depletion curve was used to translate the grid cell scale FSM outputs to fSCA (within each cell) to make it possible to assimilate the MODIS information. This is the regular way to assimilate fSCA into snow models: independently for each grid cell and snow season. The methodology is explained in Line 290, see also Line 300:
"The PBS was implemented over the fSCA ensemble over each grid cell and season independently"

**Ref.2: Line 270: Why do the authors chose a log-normal and a normal gaussian probability density function? Are just precipitation and temperature the inputs/forcing variables of the FSM2 snow model? If they are more than precipitation and temperature, how are you perturbing them in the assimilation scheme?**

Authors: We use a lognormally distributed multiplicative parameter to perturb the precipitation and a normally (Gaussian) distributed additive parameter to perturb the air temperature. A lognormal distribution, which only has positive support, is chosen for the multiplicative precipitation perturbation parameter since precipitation can't be negative, while a normal distribution, which has both negative and positive support, is chosen for the additive temperature perturbation parameter to allow for both positive and negative additive perturbations. So, aligned with the Bayesian underpinnings of data assimilation (Wikle and Berliner, 2007), we are selecting the distributions for these uncertain parameters based on physical constraints and prior knowledge. Note that these forcing perturbations are closely in line with previous applications of the PBS for snow reanalysis (e.g. Margulis et al., 2015; Cortes et al., 2015; Fiddes et al., 2019). The energy and mass balances in FSM are driven by standard hydrometeorological forcing variables; i.e. near surface air temperature, wind speed, specific humidity, precipitation, and incoming longwave and shortwave radiation. The reason that we do not perturb more forcing parameters is that by doing so we would enlarge the dimensions of the parameter space which, due to the curse of dimensionality, would make degeneracy more likely with the PBS especially since we are assimilating a relatively large number of independent observations (van Leeuwen 2009; Margulis et al., 2015). Our choice of perturbing precipitation (whose phase is controlled by air temperature) in particular is justified by the fact that precipitation bias is often the key uncertain factor controlling physically-based snow models (Raleigh et al., 2015).

New references:
Wikle and Berliner (2007), A Bayesian Tutorial for Data Assimilation, Physica D, https://doi.org/10.1016/j.physd.2006.09.017
Raleigh et al. (2015), Exploring the impact of forcing error characteristics on physically based snow simulations within a global sensitivity analysis framework, HESS, https://doi.org/10.5194/hess-19-3153-2015

**Ref.2: Line 319: How was the snow depth measured? Why not to do the comparison in term of snow depths avoiding to use a constant density value? Reading Essery (2015)FMS2 provides snow depth as an output.**

Authors: As explained in the text, we wanted to compare the snow output of the ICAR model directly, which is provided just in terms of SWE.

**Ref.2: Lines 328-342: The methodology explained here is not clear. Neither the reasoning behind nor the way SWE is compared against satellite observation. Are you using SWE measuring using remote sensing?**

Authors: We have split the paragraph in two different ones at line 330. It helps to clarify this as the SWE comparison is not related with the remote sensing part.

**Ref.2: Lines 345-381: I miss numbers supporting the statements throughout the section. Here a few examples: "Figure 2 shows how the ICAR model was able to improve the 2 m air temperature data, compared with ERA5 reanalysis", "ICAR reduces the spread of the daily precipitation errors". Moreover, I think it could be interesting to analyse a deeper when the error between observations and simulations occurs. Are they bigger in winter than in summer? Is there any dependence with the total precipitation of the hydrological year (dry or wet)? It may have large impact in your results.**

Authors: We have added numbers to the statements of the section. We agree that a deeper error assessment of ICAR should be done as it is a very new regional atmospheric model under continuous development. However, the mountains of Lebanon are not an appropriate location to do this in due to the limited data availability. As for the suggestions: i) here there is no observed information in summer (Fig. 2 and 3) and ii) it is not possible to define the dry and wet years due to the very short length of the observed series (Fig.3).

**Ref.2: Figure 3: In general, ICAR precipitation values seems to be higher than ERA5 precipitation values. However, the bias in ERA5 are positive and bigger than ICAR. How dothe authors explain that?**

Authors: Exactly, the ICAR precipitation values are higher than ERA5 values, but the difference between ERA5 and the observations is bigger than between ICAR and observations. ERA5 is too dry over the area, likely due to the lack of orographical precipitation as consequence of the smooth of the topography of the ERA5 spatial resolution.

**Ref.2: Figure 4: How do the authors explain the heterogeneous differences in the assimilation results between years? The authors gave some explanation about one of the years in Lines 399-410, however, could you give deeper explanations about the differences between years in the whole period?**

Authors: Fig.4 as well as Fig3 and 2 should be used with caution as highlighted in the text. There is a big scale missmatch between the point scale AWS information and the reanalysis. The reanalysis seems to perform better at higher elevations (Fig4 A and C), suggesting difficulties to model the precipitation partition phase, as other authors have shown over the region (Line 280). It is not possible to offer any convincing reason about the (mostly small) differences in the performance between years. The snow wind redistribution processes strongly controls the snowdepth at the point scale resolution of the AWS making comparisons complicated. In addition, previous studies have highlighted the very high snow depth spatial variability over the area as remarked in the text.

**Ref.2: Lines 421-448 and Figure 5: If I read well, here you are comparing the results of your assimilation (ICAR_assim) with the assimilated variable (Obs). This is a prove that your assimilation scheme works well, and therefore, the obtained metrics should be interpreted as**

**that. The real impact of the assimilation scheme on snow dynamics is the show in the comparison with the independent variable SWE, not assimilated during the process.**

Authors: Exactly, to clarify this point we have added the following to the text:
"...showing the potential of fSCA assimilation through the PBS in improving the ICAR SWE products."

**Ref.2: Section 4.3: All section is written as if the simulated values were a "ground truth", I would indicate some of the limitations of the performed simulations and all the sources of uncertainty and errors that are conditioning these statements.**

Authors: ICAR limitations are already described in lines 349-355. In addition, we have added the following sentences to the section 4.3 :
"ICAR_assim exhibits some limitations that should be considered. First, despite the high resolution of the reanalysis the regional nature of the simulations prevent the representation of some processes like wind or avalanche snow redistribution. In addition, there are some other sources of uncertainty involved in the development of the reanalysis, like the depletion curve, the fSCA derived from MODIS or the structural uncertainty associated with each model. However, ICAR_assim has been shown to be consistent with the few observations providing a valuable resource in the data scarce context of the Lebanese mountains.

**Ref.2: Figures 9 and 10: What does the relative area referring to (snow area over the area of the band or area of the band over all area of the mountainous ranges)? It would be interesting to see these two graphs in both mountainous ranges.**

Authors: It is already described in the text (Line 501):
"The relative area lying at each elevation compared with the total elevation over 1300 m a.s.l..."
We have added the Lebanon and Anti-Lebanon ranges SWE and accumulated water partition.

**Lines 504-508: Could you elaborate more the reasoning in this paragraph?**

Authors: We have added the following:
"This suggest that the mean peak SWE series at lower elevations could hide a large variation in mass due to the wider areas at lower elevations *where many different peak SWE values can coexist,...*"

**Ref.2: Figure 10: How do you explain that the total storage at 2800 m a.s.l. increases?**

Authors: This is because the storage at 2800m a.s.l. integrates all the surface over 2800. We have modified Fig,10 to clarify it.

**Technical comments**
**Ref.2: Figure 1. What are A and B, could the authors specify it in the body text lines 58-62 and in the figure caption.**

Authors: Fig1 legend indicates the meaning of A (WRF domain) and B (ICAR domain). We think that in this way it remains clear to readers.
The topic of Lines 58-62 do not match the atmospheric models domains.

**Ref.2: A figure with a scheme of the implementation process would help to better understand the complexity of the flow chart followed.**

Authors: We have added a schematic flowchart to the Section 3.2.2 summarizing the whole process.

**Ref.2: Figures 2, 3 and 4: It is difficult to know in which season of the year you are with the format "Days since". I propose to add actual dates in x-axis of these figures. Moreover, it is complicate to see differences between the 3 represented variables, especially in the precipitation graph. Finally, it is difficult to see what the values of the boxplot are represented, I would recommend here to change the y-axis limits, add, y-axis values and/or a grid.**

Authors: We have added the suggested changes.

---

## Author Comment (AC5) · 10 Oct 2020

Authors: We would like to thank the anonymous referee for his/her interest and the comments on our manuscript. Bellow we provide a point by point answer to the issues raised by referee #3.

**Ref.3: The paper presents an approach to downscale ERA5 reanalysis by using MODIS fSCA information. Even though the approach is not completely innovative, the research has a high relevance for the application in arid areas. Below detailed comments.**

**Ref.3: p.3 line 107:please provide here a clear statement about the objectives of the work and the innova-tive part with respect to the current literature.**

Authors: We have added the following sentence to the text:
"The objectives here are: i) to explore the potential of a methodology to develop a snowpack reanalysis over data scarce regions and ii) to describe the main snowpack dynamics over the Lebanese mountains being the first use of ICAR for this approach"

**Ref.3: Section 3.2.1. More detailed information about the processing of MODIS data need to be introduced here. Please add the new adapted linear function that the authors found by using Theia data and the explanation why it differs from the Salomonson&Appel2004.**

Authors: The equation of the linear fit is fSCA [%] = 1.23 x NDSI + 23.48. It differs from the equation of Salomonson and Appel (2004) because the calibration site is different. Salomonson and Appel (2004) obtained their relationship using Landsat-derived fSCA over Alaska, Labrador, and Siberia.

**Ref.3: How were MOD and MYD images merged? is there a cloud gap filling procedure? If the use of MYD produces a higher error why do not use only MOD?**

Authors: In fact this is what we did (see line 230).

**Ref.3: Was a validation with ground measurements conducted? this can provide an independent source of information to better quantify the accuracy of the new proposed linear relationship.**

Authors: The scarce snow depth data are already used in the other section of the manuscript. Theia Sentinel-2 snow products were extensively evaluated by Gascoin et al. (2019). For example, the comparison with automatic snow depth measurements in the Alps and Pyrenees showed that the accuracy (proportion of correct classifications) was 94 % and the kappa coefficient was 0.83.

**Ref.3: Moreover in the validation, a comparison of the new linear relationship with the one proposed by Salomonson&Appel2004is advisable to understand the advantage of the new approach.**

Authors: With the Salomonson and Appel (2004) equation we find slightly larger mean absolute error (6.2% vs 5.7%) and RMSE (12% vs. 11%) (figure below)

[Figure]

**p.9, line 320: as the density can change during the season, please justify the use of the value.**

Authors: It is true that density varies during the season. However, we had to use a fixed density value to compare the ICAR snow outputs with the snow depth observations at the AWS. Fayad et al., 2017a showed that such value is the mean density of the snowpack in the area. We showed the uncertainity caused by this value with a sensitivity analysis in Fig. 4

---

## Author Response (AR2)

We appreciate Ref1 comments, they are a good contribution to improve the overall understanding of the document. Review comments below are reproduced in blue and responses are in black.

The authors examined a methodology to obtain snowpack reanalysis using ICAR and data assimilation methodology over data scarce regions. The work is valued since the authors take the advantage of using atmospheric reanalysis data together with remote sensing and integrating techniques of downscaling and data assimilation. There are number of analysis in comparison with ground truth (AWS data sets) and remote sensed observations (fSCA of MODIS) in the study. In this regard, the study addresses an interesting topic which fits well with the scope of HESS and it is also an important contribution to the snow science especially for the mountainous regions where there is a data scarcity.

The authors replied a number of comments in the previous review step which has already added value to the study. There are still couple of important aspects that need to be refined so that the possible impact of the study would further increase.

Each process has its own contribution with different advantages, WRF improves the resolution compared to ERA5 and ICAR improves WRF results even more with computational efficiency. It would be remarkable to see the contribution of ERA5, WRF and ICAR in temperature and precipitation reanalysis separately with simply showing the same error analysis provided in Figure 3 and Figure 4. Even if ICAR will not improve WRF in terms of statistical performance it is still required for any analyzes/processes need a better resolution.

We have included the WRF outputs in the Figure 3 and 4, for a better understanding of the contribution of each process in the workflow. However, such comparison should be taken with care. The lack of ground-based data and short length of the AWS series makes difficult to extract conclusions in the comparison of the gridded products, as the ICAR/WRF/ERA5 sub-cell variability is not properly represented by the AWS due to the very complex topography.

According to the text, ICAR snowpack reanalysis data is acquired as a result of ICAR output. This step should be indicated in the flow chart as an output (in Figure 2). Then, the authors improve ICAR snowpack reanalysis since these are not very consistent with the observations (Observed and ICAR in Figure 5), so in the next step they prefer to implement data assimilation using satellite snow cover data. The authors should clarify this part and refine the role of Flexible Snow Model (FSM) in the study (which seems also to be the concern of reviewers previously); it looks as if there are two snowpack data sets: one as a direct output of ICAR reanalysis (ICAR in Figure 5) and the other is an output of FSM where the forcing variables are ICAR precipitation and temperature reanalysis data without data assimilation (which is not provided in the text or figures). The assimilation of MODIS fSCA is applied to the later as indicated, ICAR_assim, in Figure 2 and Figure 5. If this is so, to make a fair comparison, my suggestion is to use FSM results without data assimilation instead of ICAR snowpack reanalysis output (ICAR) in Figure 5 and also in the statistical performance analysis. This will also change the conceptualization in the study and objectives since ICAR_assim is not a direct output of ICAR snowpack reanalysis as authors emphasize that it is an ensemble output of FSM as a result of perturbation of ICAR reanalysis forcings.

ICAR snowpack reanalysis (ICAR_assim) is generated fusing the fSCA MODIS retrievals with FSM outputs forced by perturbed ICAR surface variables. It is already pointed in Fig2, we can not add any new step there. The use of a decoupled snowpack model (FSM) is mandatory, as an ensemble of snowpack simulations is required to implement the PBS. We have added the FSM outputs forced by ICAR unperturbed surface variables in the Figure5 as Ref1 requested. FSM and ICAR performs similarly during most of the situations, with differences by the end of some seasons where FSM has exhibited a faster decline. As the uncertainty associated to using different snowpack models, is lower than the uncertainty induced by the forcing, we hypothesize that the differences between ICAR_Noah and ICAR_FSM are mainly as consequence of the precipitation phase partitioning parametrization used in this work, that is less robust than the Thompson microphysics scheme implemented in ICAR. These difficulties on simulate the precipitation phase are typically found in the mild climates, where the spring snowfalls occur close to the zero-isotherm elevation.

We have included the following paragraph in the text, to explain the contribution of FSM to ICAR_assim:

[*The use of FSM to generate the ensemble of simulations, introduced some uncertainties in the workflow. Some water years showed earlier snow melts using the FSM forced by ICAR, compared with the ICAR snow outputs. As the uncertainty of the snow models associated to the forcing is higher*

*than the uncertainty associated by the use of different model parameterizations and model structures (Günther et al., 2019), we hypothesize that such differences were caused by the differences in the precipitation phase partitioning, which is challenging to simulate in the areas that remain close to 0 °C during the snow season (Fayad and Gascoin, 2020). The lack of spring snowfalls in some years may have deep implications in the snowpack simulation that are not limited to its effect in the mass balance and the releasing of latent heat by refreezing the liquid precipitation. It leads to lower albedos, which combined with the high short-wave radiation of Lebanon due to its latitude causes earlier snow melts. However, such discrepancies are greatly minimized in ICAR_assim, by the assimilation of the fSCA retrievals.* ]

There is an important impact of PBS assimilation on the results but this is rather unrealistic for some cases where there is almost no snow water equivalent before assimilation (Figure 5). Providing FSM results with forcing of ICAR before and after data assimilation as suggested above might better explain this remarkable contribution of data assimilation by PBS. If the description above (on direct ICAR snowpack output and FSM snowpack output without data assimilation) is not accurate then the authors should better explain the reasoning behind such a spectacular change in SWE with data assimilation through the use of fSCA of MODIS. Since there is a remark on NDSI reformulation using Theia in fSCA analysis to improve the results compared to the ordinary equation, it should also be important to deal with cloud removal process and give more explanation in section 3.2.1.

The use of FSM forced by the unperturbed surface meteorological variables of ICAR does not improve the snow simulations compared to ICAR_Noah, actually the implementation of FSM suffered from a less robust precipitation phase partitioning scheme, as explained above. It should be noticed that the comparison between the 1km grid scale of the simulations and the AWS exhibited differences due to the very variable nature of the snow at the point scale of the AWS, which is obviously not represented by the 1km resolution. The impact of smoother-based data assimilation schemes on snow simulations is well-known in the literature. Specifically, the use of PBS to assimilate fSCA observations into a snowpack model achieved similar or even better performances comparing with in-situ observations in the Californian Sierra Nevada (Margulis et al., 2016). Similar improvements in the simulations were found after the implementation of the PBS to assimilate fSCA retrievals in the Andes (Cortés et al., 2016), Switzerland (Fiddes et al., 2019), and Svalbard (Aalstad et al., 2018). We would like to emphasize that these implementations of the PBS and other smoother schemes work by correcting the forcing (mainly biases in solid precipitation) for a given water year through a Bayesian updating of the ensemble of modelled fSCA trajectories using the observed fSCA trajectory from MODIS. This, in turn, updates the entire SWE trajectory in a batch (i.e. for the entire water year, rather than sequentially) through the posterior forcing time series. Such updates can be considerable. For example, if the prior mean melts out too early relative to the observed fSCA, then the posterior mean SWE will typically be higher than the prior mean since it is only the trajectories with considerably higher snowfall that can reproduce the observed fSCA depletion. The converse case occurs when the prior mean melts out too late. This hopefully explains the "spectacular change" in SWE that can occur with the PBS. Such an update would not be possible with the widely used (for snow DA) filtering algorithms which directly update SWE sequentially in time, since with filters the information from observations can not propagate backwards in time. We did not follow any specific procedure for the cloud removal, as the cloud masks are already processed and included in the remote sensing products used in this study (MODIS and Theia snow).

References:

Aalstad, K., Westermann, S., Schuler, T. V., Boike, J. and Bertino, L.: Ensemble-based assimilation of fractional snow-covered area satellite retrievals to estimate the snow distribution at Arctic sites, Cryosphere, 12(1), 247–270, doi:10.5194/tc-12-247-2018, 2018.

Cortés, G., Girotto, M. and Margulis, S.: Snow process estimation over the extratropical Andes using a data assimilation framework integrating MERRA data and Landsat imagery, Water Resour. Res., 52(4), 2582–2600, doi:10.1002/2015WR018376, 2016.

Fayad, A. and Gascoin, S.: The role of liquid water percolation representation in estimating snow water equivalent in a Mediterranean mountain region (Mount Lebanon), Hydrol. Earth Syst. Sci., 24(3), 1527–1542, doi:10.5194/hess-24-1527-2020, 2020.

Fiddes, J., Aalstad, K. and Westermann, S.: Hyper-resolution ensemble-based snow reanalysis in mountain regions using clustering, Hydrol. Earth Syst. Sci., 23(11), 4717–4736, doi:10.5194/hess-23-4717-2019, 2019.

Günther, D., Marke, T., Essery, R. and Strasser, U.: Uncertainties in Snowpack Simulations—Assessing the Impact of Model Structure, Parameter Choice, and Forcing Data Error on Point-Scale Energy Balance Snow Model Performance, Water Resour. Res., 55(4), 2779–2800, doi:10.1029/2018WR023403, 2019.

Margulis, S. A., Cortés, G., Girotto, M. and Durand, M.: A landsat-era Sierra Nevada snow reanalysis (1985-2015), J. Hydrometeorol., 17(4), 1203–1221, doi:10.1175/JHM-D-15-0177.1, 2016.

The authors thank Ref.2 for their valuable comments. We provide a point by point answer bellow. Review comments below are reproduced in blue and responses are in black.

*Line 51: Please provide a reference to this new statement.*

Authors: We have moved the (López-Moreno and García-Ruiz 2004) reference to include this new statement.

*Line 325: How was the snow depth measured in the AWS? I asked before and I think I did not see the answer.*

It's a Campbell SR50A acoustic gauge. We have added the model of the sensor in the text:

"[...]information derived from a Campbell SR50A acoustic gauge at the three AWS."

*Figures 3, 4, 5, 6: I appreciate the authors changed the format "Date since" for actual values. However, I still miss the ticks in the x-axis with the exact location of the dates in some of the figures (You have done that in the y-axis or in x-axis of Figure 6). In addition, more dates (smaller time scales: months) would be appreciated for those sites with less analysed years. As I commented before a grid would help to interpret these figures.*

We have added tick marks to the end/begging of the years as well as new reference lines to these figures. We have preferred to keep the same data scales in all the plots in the benefit of the consistency.

*Figure 10: Why did the red dotted line change in this version of the manuscript?*

The difference was caused by an inconsistency in our framework caused by applying the suggested changes by the previous review. It is fixed now using a more robust routine, thanks.

*Supplementary 1: What does the colour scale represent? I understand it is number of pixels that fall within each grid. However, if that the case they seem less than 40% of 58400, but maybe I am wrong. In any case, I would appreciate if the authors included the metrics (RMSE and MAE) in the figure. Moreover, could the authors also add the value of determination coefficient for both linear fitting? I think is a better metric in this case. I would also recommend adding the same plot for the validation period.*

The colour scale represent the relative density of the scatter plot in a log10 scale (added to the caption). The colour scale was designed to improve the interpretation of the figure. The RMSE and MAE metrics are since last version included in the 4.2 section, we apologize but the routine used to estimate the linear fit was primally designed to optimize the linear and non-linear functions, for which the R2 would not be relevant. To obtain the r2 it would be necessary to develop deep modifications in the code.

---

## Author Response (AR3)

We have answered (in black) the Referee #2 comments bellow *(in blue)*

*I would like to thank the authors their answer to my questions. I still find some some technical issues that should be addressed before publication:*
*Figures 3, 4 and 5: Thanks for the horizonal lines, the vertical ones would been appreciated too. I think you should differentiate the panels in these figrues, for instance using letters (a, b, c) and specify in the caption what they refer each of them.*
We have included vertical lines and ABC titles to the plots that correspond with the names of the AWSs in table 1.

*Figure 5: In the first panel the legend does not allow to see the actual lines.*
Corrected, thanks

*Figure 11 (initially Figure 10): The figure has changed completely. I understand that the inconsistency you mentioned in your answer refers not only to the red dot line but also to the Total Water Storage simulation. Is that correct?*
Exactly, as the error came from the way we read the DEM to discriminate the different elevation bands.

*Supplementary figure: Yes, I knew that the metrics, RMSE and MSE, are in section 4.2. My comment suggested to include them also in the figure to have the info here too. Moreover, I suggest to change the unit of fSCA and NDSI to make them coherent with the ones used in the equation in line 403.*
Included.